# Investigation of the Antiremodeling Effects of Losartan, Mirabegron and Their Combination on the Development of Doxorubicin-Induced Chronic Cardiotoxicity in a Rat Model

**DOI:** 10.3390/ijms23042201

**Published:** 2022-02-16

**Authors:** Marah Freiwan, Mónika G. Kovács, Zsuzsanna Z. A. Kovács, Gergő Szűcs, Hoa Dinh, Réka Losonczi, Andrea Siska, András Kriston, Ferenc Kovács, Péter Horváth, Imre Földesi, Gábor Cserni, László Dux, Tamás Csont, Márta Sárközy

**Affiliations:** 1MEDICS Research Group, Department of Biochemistry, Albert Szent-Györgyi Medical School, University of Szeged, H-6720 Szeged, Hungary; marah.mf.94@gmail.com (M.F.); kovacs.monika.gabriella@med.u-szeged.hu (M.G.K.); kovacs.zsuzsanna@med.u-szeged.hu (Z.Z.A.K.); szucs.gergo@med.u-szeged.hu (G.S.); dinhhoaqa@gmail.com (H.D.); losonczireka1997@gmail.com (R.L.); csont.tamas@med.u-szeged.hu (T.C.); 2Interdisciplinary Center of Excellence, University of Szeged, H-6720 Szeged, Hungary; 3Department of Laboratory Medicine, Albert Szent-Györgyi Medical School, University of Szeged, H-6720 Szeged, Hungary; siska.andrea@med.u-szeged.hu (A.S.); foldesi.imre@med.u-szeged.hu (I.F.); 4Synthetic and Systems Biology Unit, Biological Research Centre, Eötvös Loránd Research Network, H-6726 Szeged, Hungary; kriston.andras@single-cell-technologies.com (A.K.); kovacs.ferenc@single-cell-technologies.com (F.K.); peter.horvath@brc.hu (P.H.); 5Single-Cell Technologies Ltd., H-6726 Szeged, Hungary; 6Institute for Molecular Medicine Finland (FIMM), University of Helsinki, FIN-00014 Helsinki, Finland; 7Department of Pathology, Albert Szent-Györgyi Medical School, University of Szeged, H-6720 Szeged, Hungary; cserni.gabor@med.u-szeged.hu; 8Muscle Adaptation Group, Department of Biochemistry, Albert Szent-Györgyi Medical School, University of Szeged, H-6720 Szeged, Hungary

**Keywords:** onco-cardiology, doxorubicin-induced chronic cardiotoxicity, heart failure, cardiac fibrosis, diastolic dysfunction, angiotensin II receptor blocker, beta-3 adrenoceptor agonist, cardiac inflammation, TGF-β/SMAD signaling pathway, sarcoendoplasmic reticulum calcium ATPase 2a

## Abstract

Despite the effectiveness of doxorubicin (DOXO) as a chemotherapeutic agent, dose-dependent development of chronic cardiotoxicity limits its application. The angiotensin-II receptor blocker losartan is commonly used to treat cardiac remodeling of various etiologies. The beta-3 adrenergic receptor agonist mirabegron was reported to improve chronic heart failure. Here we investigated the effects of losartan, mirabegron and their combination on the development of DOXO-induced chronic cardiotoxicity. Male Wistar rats were divided into five groups: (i) control; (ii) DOXO-only; (iii) losartan-treated DOXO; (iv) mirabegron-treated DOXO; (v) losartan plus mirabegron-treated DOXO groups. The treatments started 5 weeks after DOXO administration. At week 8, echocardiography was performed. At week 9, left ventricles were prepared for histology, qRT-PCR, and Western blot measurements. Losartan improved diastolic but not systolic dysfunction and ameliorated SERCA2a repression in our DOXO-induced cardiotoxicity model. The DOXO-induced overexpression of *Il1* and *Il6* was markedly decreased by losartan and mirabegron. Mirabegron and the combination treatment improved systolic and diastolic dysfunction and significantly decreased overexpression of *Smad*2 and *Smad3* in our DOXO-induced cardiotoxicity model. Only mirabegron reduced DOXO-induced cardiac fibrosis significantly. Mirabegron and its combination with losartan seem to be promising therapeutic tools against DOXO-induced chronic cardiotoxicity.

## 1. Introduction

Cancer and cardiovascular diseases (CVDs) are the leading causes of morbidity and mortality worldwide [1,2]. Due to advances in early diagnosis and treatment of cancer patients, long-term cancer survivors are one of the largest growing populations accessing the healthcare system [3,4]. After recurrent malignancies, CVDs are the second leading cause of morbidity and mortality in cancer survivors [3]. Cancer therapies, particularly chemo and radiotherapy, have many recognized side effects on the cardiovascular system [3,5,6]. In early and late chronic stages, chemotherapy-induced cardiotoxicity commonly manifests in decreased left ventricular ejection fraction (LVEF), leading to heart failure (HF) symptoms [3,6].

Anthracyclines, including doxorubicin (DOXO), are essential drugs in chemotherapeutic regimens in different cancers, such as leukemias, lymphomas, soft tissue sarcomas, and solid malignancies (i.e., breast, ovary, prostate, stomach, thyroid, liver, and small cell lung cancers) [7,8]. Although anthracyclines are effective and commonly used chemotherapeutic agents, their application could be limited by the dose-dependent development of cardiotoxicity. Anthracyclines-induced cardiotoxicity can manifest in acute, early chronic, and late chronic forms. Acute toxicity is usually reversible and predominantly presents supraventricular arrhythmias, transient left ventricular dysfunction, and electrocardiographic changes in less than 1% of patients immediately after treatment [6]. Notably, acute cardiac dysfunction may lead to early or late chronic cardiotoxicity characterized by systolic dysfunction [9]. Early chronic cardiotoxic signs occur within the first year of treatment, while late effects present after several years (median of 7 years after treatment) [3,10,11]. In the case of DOXO, the risk for developing chronic cardiotoxicity is 5% at a cumulative dose of 400 mg/m^2^, 26% at a dose of 550 mg/m^2^, and 48% at a dose of 700 mg/m^2^ in humans [6]. Patients under 18 or over 65 years, suffering from cardiovascular comorbidities such as hypertension, left ventricular hypertrophy, coronary artery disease, diabetes mellitus, or prior radiation exposure, are at higher risk for developing DOXO-induced chronic cardiotoxicity [3,6].

The basic mechanisms underlying DOXO-induced chronic cardiotoxicity have not yet been fully understood. In cancer cells, DOXO was shown to bind to topoisomerase-2α, causing deoxyribonucleic acid (DNA) double-strand break and cell death [12,13]. In cardiomyocytes, DOXO was reported to target topoisomerase-2β, also leading to DNA double-strand breaks and the death of cardiomyocytes. DOXO-bound topoisomerase-2β can bind to promoters of antioxidative genes and peroxisome proliferator-activated receptor-gamma coactivator 1 (PGC1), which are needed for the expression of antioxidant enzymes and the elements of the mitochondrial electron transport chain [14]. Thus, topoisomerase-2β may be able to account for the three hallmarks of DOXO-induced cardiotoxicity, including (i) cardiomyocyte death mainly by apoptosis, (ii) generation of reactive oxygen and nitrogen species (ROS/RNS), and (iii) mitochondrial damage [12,13]. Another accepted theory is that DOXO forms an anthracycline-iron complex, which then induces lipid peroxidation, protein oxidation, and DNA damage by ROS production that results in contractile impairment, irreversible myocardial damage, and fibrosis [7,15]. At the same time, other mechanisms have been proposed, such as tissue inflammation, extracellular matrix remodeling, myofilament dysfunction, and disturbance in intracellular calcium ion (Ca^2+^) homeostasis [7,15].

Although DOXO effectively kills tumor cells, there is currently no sufficiently effective agent to prevent or treat DOXO-induced chronic cardiotoxicity without diminishing antitumor effects of DOXO or promoting secondary malignancy [3]. The renin-angiotensin-aldosterone system (RAAS) was reported to be overactivated in cardiovascular pathologies, including hypertension, cardiac hypertrophy, and heart failure leading to elevated nitro-oxidative stress, inflammation, apoptosis, and fibrosis [16]. Among the inhibitors of RAAS overactivation, angiotensin-II receptor blockers (ARBs) are widely used drugs to prevent the progression of chronic heart failure in various comorbidities [17,18]. ARB losartan showed cardioprotective effects against experimental DOXO-induced cardiotoxicity [19,20]. Indeed, based on the results of clinical trials, inhibition of RAAS overactivation with ARBs has also shown beneficial effects on the development of DOXO-induced chronic cardiotoxicity [13].

The beta-3 adrenoreceptor (β3AR) agonist mirabegron is used in urology to treat hyperactive bladder syndrome [21]. In preclinical models, the β3AR agonists attenuated cardiac fibrosis and improved cardiac contractility via coupling of β3AR to the eNOS/cGMP pathway in cardiomyocytes [22,23,24]. Moreover, the antioxidant effects of the β3AR signaling and the down-regulation of the angiotensin II type 1 receptor (AT1) in response to β3AR stimulation may protect the heart from elevated nitro-oxidative stress and the consecutive pro-inflammatory and fibrotic processes [23,25,26,27,28]. Our group recently showed moderate antifibrotic effects of mirabegron in a rat model of uremic cardiomyopathy independently of the β3AR/eNOS pathway [29]. Bundgaard reported that mirabegron significantly increased LVEF in a subset of patients with less than 40% starting LVEF compared to patients given placebo [30]. This result may suggest that mirabegron could have beneficial effects on heart failure with reduced ejection fraction (HFrEF). Indeed, the antiremodeling effects of the β3AR agonist mirabegron are being investigated in HFrEF patients in clinical trials (trial numbers: NCT03926754 and NCT02775539). However, the antiremodeling effects of mirabegron have not been studied in DOXO-induced chronic cardiotoxicity. Therefore, in our present study, we aimed at investigating and comparing the potential antiremodeling effects of the widely-used ARB losartan, the β3AR agonist mirabegron, and their combination on the development of DOXO-induced chronic cardiotoxicity in a rat model.

## 2. Results

Our present study aimed to use a DOXO-induced chronic cardiotoxicity model presenting similar cardiac pathology to those seen in tumor survivor patients treated with DOXO. The treatments with losartan, mirabegron, and their combination started after the DOXO-administration, similar to the conventionally scheduled heart failure therapeutic regimens in clinical practice [6,31]. Our experimental protocol is presented in Figure 1.

Altogether nine animals died in the DOXO groups (*n* = 1 in the DOXO-only group at week 4, *n* = 2 in the losartan treated DOXO group (one animal at week 4 before the start of losartan treatment and one animal 1 day before the termination at week 9), *n* = 4 in the mirabegron-treated DOXO group (one animal was excluded and terminated earlier due to its poor echocardiographic result and body weight at week 4, one animal before the start of mirabegron treatment due to abdominal ulceration at week 4 and two animals during the mirabegron treatment at weeks 7 and 8), and *n* = 2 in the losartan plus mirabegron-treated DOXO group (one animal was excluded and terminated earlier due to its poor echocardiographic results at week 4 and one animal during the losartan plus mirabegron treatment at week 6)).

### 2.1. Early Echocardiographic Signs of Systolic Dysfunction Developed in the DOXO Groups before Starting the Treatments at Week 4

Echocardiography was performed at week 4 to assess the effects of DOXO on cardiac morphology and function before starting the treatments with losartan, mirabegron, and their combination (Figure 1, Table 1). There were no significant differences in most of the measured morphologic and functional parameters between the DOXO and control groups (Table 1). Only the left ventricular end-systolic diameter was significantly higher in the DOXO groups compared to the control group indicating an early sign of DOXO-induced chronic cardiotoxicity assessed by echocardiography (Table 1). Additionally, fractional shortening, diastolic septal wall thickness, and systolic anterior wall thickness showed a statistically non-significant decrease in the DOXO groups compared to the control group (Table 1).

### 2.2. DOXO-Treated Groups Presented Lower Body Weight and Higher Serum Cholesterol Levels Irrespective of the Treatments with Losartan, Mirabegron, and their Combination at Week 9

There was no significant difference in body weight before the start of the DOXO administration between the groups (Table 2). After the last cycle of DOXO administration (i.e., week 0, Figure 1), the body weight was significantly lower of the DOXO-treated animals compared to the controls (Table 2). Nine weeks after the last cycle of DOXO administration, the body weight was significantly lower in the DOXO groups compared to the control group irrespective of treatments with losartan, mirabegron, or their combination (Table 2).

At week 9, there was no significant difference in routine renal functional parameters, including serum carbamide and creatinine levels, between the groups (Table 2). At week 9, serum cholesterol and triglyceride levels, and blood pressure were measured as cardiovascular risk factors. Interestingly, serum cholesterol levels were significantly higher in all DOXO groups, irrespective of treatments (Table 1). Notably, serum cholesterol levels showed a trend of decreasing in response to losartan (*p* = 0.07) and mirabegron (*p* = 0.08) compared to the DOXO-only group (Table 2). Serum triglyceride levels were significantly higher in the DOXO-only group compared to the control group (Table 2). Mirabegron significantly reduced the serum triglyceride level compared to the DOXO-only group. However, losartan or the combination treatment failed to markedly improve the serum triglyceride levels compared to the DOXO-only group (Table 2). There were no significant differences in the systolic, diastolic, and mean arterial blood pressure values between the DOXO-only and the control groups (Table 2). Due to its antihypertensive effects, losartan markedly decreased the systolic, diastolic, and mean arterial blood pressure compared to the values of the control or the DOXO-only groups (Table 2). Mirabegron did not significantly change the blood pressure parameters compared to the control or DOXO-only groups (Table 2). Notably, the combination treatment showed a trend toward a decrease in the systolic (*p* = 0.13), diastolic (*p* = 0.08), and mean (*p* = 0.07) arterial blood pressure compared to the DOXO-only group (Table 2).

### 2.3. Echocardiographic Signs of the DOXO-Induced Chronic Cardiotoxicity Were Alleviated by Mirabegron and the Combination Treatment but Not by Losartan at Week 8

Echocardiography was performed at week 8 to assess the effects of losartan, mirabegron, or their combination on the DOXO-induced pathologic changes in cardiac morphology and function (Figure 1, Figure 2 and Figure 3, Table 3). The systolic septal, posterior, anterior, inferior, and diastolic septal wall thicknesses were significantly smaller in the DOXO-only group compared to the control group (Figure 2a–d, Table 3). There was no significant change in the left ventricular end-diastolic diameter; however, the left ventricular end-systolic diameter was markedly increased in the DOXO-only group compared to the control group (Figure 2e,f). Consequently, fractional shortening and ejection fraction was significantly reduced in the DOXO-only group compared to the control group, indicating the development of systolic dysfunction in response to DOXO (Figure 2, Table 3). Additionally, another systolic parameter, the isovolumic contraction time, was significantly prolonged in the DOXO-only group compared to the control group (Table 3). There were no significant differences in the diastolic posterior, anterior and inferior wall thicknesses between the DOXO-only and control groups. Notably, the heart rate tended to decrease (*p* = 0.07) in the DOXO-only group compared to the control group (Table 3).

Comparing the losartan-treated DOXO-group to the control group, the systolic septal and anterior wall thicknesses, fractional shortening, and ejection fraction were significantly reduced, and the left ventricular end-systolic diameter was significantly increased, similar to the results of the DOXO-only group (Table 3, Figure 2). It should be mentioned that the isovolumic contraction time was significantly reduced by losartan compared to the DOXO-only group (Table 2). There were no significant differences in the diastolic septal, posterior, anterior, and systolic posterior and inferior wall thicknesses, left ventricular end-diastolic diameter, and heart rate between the losartan-treated DOXO and DOXO-only or control groups (Table 2, Figure 3).

In response to mirabegron, the systolic septal, anterior and inferior wall thicknesses were significantly increased compared to those in the DOXO-only group (Table 3, Figure 2). Accordingly, there were no significant differences in the left ventricular end-systolic diameter, fractional shortening, ejection fraction, and isovolumic contraction time between the mirabegron-treated DOXO and control groups (Table 3, Figure 2). In response to mirabegron, the heart rate and the other measured wall thicknesses were not significantly different from those in the DOXO-only or control groups (Table 3).

There were no significant differences in most wall thicknesses, left ventricular end-systolic and end-diastolic diameters, fractional shortening, ejection fraction, and isovolumic contraction time in the losartan plus mirabegron-treated DOXO group compared to the control group or the DOXO-only group (Table 3, Figure 2). However, it should be mentioned that fractional shortening and ejection fraction showed decreasing tendencies (*p* = 0.13 and *p* = 0.14, respectively) in the losartan plus mirabegron-treated group compared to the control group. Accordingly, the isovolumic contraction time tended to increase in the losartan plus mirabegron-treated group compared to the control group (*p* = 0.06 using unpaired *t*-test). Notably, systolic inferior wall thickness and heart rate were significantly reduced, and the systolic posterior and diastolic septal wall thicknesses showed a tendency of decrease in the losartan plus mirabegron-treated DOXO group compared to those in the control group (Table 3, Figure 2).

In the DOXO-only group, the septal mitral annulus velocity e’ was significantly smaller, and the early flow velocity E was unchanged, leading to markedly higher E/e’ compared to those of the control group, indicating diastolic dysfunction (Figure 3a–d). Another parameter of the diastolic function, the isovolumic relaxation time, was significantly prolonged in the DOXO-only group compared to the control group (Figure 3e). In response to losartan, the E, e’, and their ratio were not significantly different from those of the control or DOXO-only groups (Figure 3a–d). Notably, losartan significantly reduced the isovolumic relaxation time compared to the DOXO-only group (Figure 3e). The treatment by mirabegron did not affect the E significantly, whereas it markedly increased the e’ and reduced the E/e’ compared to the values of the DOXO-only group (Figure 4e). In contrast, the isovolumic relaxation time remained significantly longer in the mirabegron-treated DOXO group than in the control group (Figure 3e). In response to losartan plus mirabegron treatment, E was markedly reduced compared to the control group (Figure 3a,c). The treatment by losartan plus mirabegron significantly increased the e’ and reduced the E/e’ compared to the values of the DOXO-only group (Figure 3b,d). The isovolumic relaxation time was not significantly different between the losartan plus mirabegron-treated DOXO groups and the control or DOXO-only groups (Figure 3e).

### 2.4. DOXO-Indued Heart Weight Loss Was Alleviated by Mirabegron at Week 9

At weeks 9, hearts, lungs, and tibias were isolated, then left and right ventricles were separated, and the organ weights and tibia lengths were measured (Table 4). Tibia lengths were not significantly different between the groups (Table 4). However, the tibia lengths in the DOXO-only (*p* = 0.081) and losartan-treated DOXO groups (*p* = 0.065) showed a trend to a decrease compared to the control group (Table 4). Heart weight, as well as left and right ventricular weights, were significantly lower in the DOXO-only group compared to those of the control group (Table 4). Right ventricular weight was not significantly different between the losartan-treated DOXO and control groups (Table 4). In response to losartan, the heart weight and left ventricular weight remained significantly smaller as compared to those of the control group (Table 4).

In response to mirabegron, heart weight, as well as left and right ventricular weights, were not markedly different from the values of the control group (Table 4). In contrast, the combination of losartan and mirabegron resulted in significantly smaller heart weights and left and right ventricular weights than those of the control group (Table 4). Lung weight showed a decreasing trend in the DOXO-only group compared to the control group (*p* = 0.13). Lung weights in the DOXO groups treated with losartan, mirabegron and their combination were not significantly different from those of the control group (Table 4).

### 2.5. DOXO-Indued Cardiac Fibrosis Was Significantly Reduced Only by Mirabegron at Week 9

To characterize the potential anti-remodeling effects of losartan, mirabegron, or their combination in DOXO-induced cardiotoxicity, cardiomyocyte cross-sectional areas were measured on hematoxylin-eosin-stained sections, and fibrosis was quantified on picrosirius red and fast green-stained sections (Figure 4a–d).

There was no significant difference in the cardiomyocyte cross-sectional areas between the DOXO-only and control groups (Figure 4a,b). Interestingly, losartan significantly reduced the cardiomyocyte cross-sectional area compared to the control group (Figure 4a,b). Mirabegron and the combination of losartan and mirabegron did not significantly affect the cross-sectional areas compared to those of the control or DOXO-only groups (Figure 4a,b).

In the DOXO-only group, the interstitial collagen content of the left ventricles was significantly increased compared to the control group, indicating the development of LV fibrosis (Figure 4c,d). In response to mirabegron, the LV collagen content was significantly smaller than the DOXO-only group (Figure 4c,d). In response to losartan or the combination treatment, the LV collagen content was not significantly different from the control or the DOXO-only group (Figure 4c,d). Notably, the LV collagen content showed a trend to a decrease in response to losartan plus mirabegron compared to the DOXO-only group (*p* = 0.085, Figure 4c,d).

LV expressions of the collagenases matrix metalloprotease-2 (*Mmp2*) and matrix metalloprotease-9 (*Mmp-9*) were significantly increased in the DOXO-only group compared to the control group (Table 5). In response to losartan, the *Mmp9* remained significantly overexpressed, and the *Mmp2* expression was not significantly different from the control group (Table 5).

In response to mirabegron, *Mmp2* and *Mmp9* expressions were not markedly different from those of the control group (Table 5). The combination of losartan and mirabegron reduced the *Mmp9* expression significantly and resulted in a decreasing tendency in *Mmp2* expression as compared to the DOXO-only group (Table 5).

The heart failure markers natriuretic peptides A and B (*Nppa* and *Nppb*, respectively) were significantly overexpressed in the DOXO-only group and remained significantly higher in the mirabegron-treated DOXO group compared to the control group (Table 5). Losartan significantly reduced *Nppa* overexpression, and the combination of losartan plus mirabegron markedly decreased *Nppb* expression compared to the DOXO-only group (Table 5).

The left ventricular expression of the apoptotic marker *Bax* showed a trend to increase in the DOXO-only group compared to the control group (*p* = 0.066) (Table 5). There was no significant difference in the *Bax* expression between the control and the losartan, mirabegron, or losartan plus mirabegron-treated DOXO groups (Table 5). There was no significant difference in the left ventricular expression of the antiapoptotic *Bcl2* between the groups (Table 5). Interestingly, the *Bax/Bcl2* ratio was significantly increased in the DOXO-only group compared to the control group (Table 5). Only mirabegron, but not losartan or the combination treatment, significantly decreased the *Bax/Bcl2* ratio compared to the control group (Table 5). These results are in accordance with the heart weight, left ventricular weight, and fibrosis results (Table 4 and Figure 4).

### 2.6. Overexpression of Smad2 and Smad3 Were Ameliorated by Treatments with Losartan, Mirabegron and their Combination in DOXO-Indued Chronic Cardiotoxicity

To further characterize the potential antiremodeling effects of losartan, mirabegron and their combination, left ventricular expression of several elements of the TGF-β/SMAD fibrotic pathway was measured at the transcript or the protein level in DOXO-induced chronic cardiotoxicity (Figure 5a–f). Left ventricular expressions of the connective tissue growth factor (*Ctgf*), transforming growth factor-beta receptor type II (TGFβRII), mothers against decapentaplegic homolog 2 and 3 (*Smad2* and *Smad3*), and collagen 1a1 (*Col1a1*) were significantly increased in the DOXO-only group compared to those of the control group (Figure 5a,c–f).

There was no significant difference in the left ventricular mRNA expression of transforming growth factor-beta (*Tgfb*) between the groups (Figure 5b). In response to losartan, the left ventricular expressions of *Ctgf*, TGFβRII, *Smad3*, and *Col1a1* were not significantly different from those of the control group (Figure 5a,c,e,f). Moreover, the *Smad2* overexpression was significantly decreased by losartan compared to the DOXO-only group (Figure 5d). In response to mirabegron, left ventricular expressions of *Ctgf* and TGFβRII remained significantly higher than in the control group (Figure 5a,c). *Smad2* and *Smad3* overexpression were significantly reduced by mirabegron compared to the DOXO-only group (Figure 5d,e). There was no significant difference in the *Col1a1* expression between the mirabegron-treated DOXO and control groups (Figure 5f). In response to losartan plus mirabegron, the expressions of *Ctgf*, *Tgfb*, TGFβRII, and *Col1a1* were not significantly different from those of the control group (Figure 5a–c,f). In addition, the combination of losartan plus mirabegron significantly reduced the *Smad2* and *Smad3* expressions compared to those of the DOXO-only group (Figure 5d,e).

### 2.7. DOXO-Indued Repression of SERCA2a Was Ameliorated by Losartan at Week 9

Decreased sarcoendoplasmic reticulum calcium ATPase 2a isoform (SERCA2a) expression and disturbed calcium homeostasis are related to impaired diastolic relaxation and reduced systolic function [32]. In our present study, the left ventricular expression of the SERCA2a protein was significantly decreased in the DOXO-only group compared to the control group (Figure 6a). Losartan markedly increased the left ventricular SERCA2a expression compared to the DOXO-only group (Figure 6a). In cases of the mirabegron and the combination-treated groups, the SERCA2a expression showed a trend to decrease compared to the control group (*p* = 0.062 and *p* = 0.054, respectively) (Figure 6a).

### 2.8. Left Ventricular β3AR Protein Levels were Decreased in all DOXO-Treated Groups Irrespective of Treatments with Losartan, Mirabegron, and their Combination

Myocardial overexpression of the β3AR and dysfunction in the eNOS-mediated pathways were reported in heart failure of different etiologies [21]. In contrast, the β3AR protein level was significantly decreased in the DOXO-treated groups irrespective of treatments with losartan, mirabegron, and their combination, compared to the control group (Figure 6b). There were no significant differences in the eNOS and p-eNOS levels and the p-eNOS/eNOS ratios among the groups (Figure 6c–e).

### 2.9. Left Ventricular Expression of the Inducible Nitric Oxide Synthase was Increased in All DOXO-Treated Groups Irrespective of Treatments with Losartan, Mirabegron, and their Combination

The left ventricular expressions of neuronal (*Nos1*) and inducible (*Nos2*) nitric oxide synthase were measured at the transcript level by qPCR (Table 6). NOS1 is localized in the myocardial sarcoplasmic reticulum and modulates cardiac function and intracellular Ca^2+^ fluxes [33]. However, no significant difference was found in the left ventricular *Nos1* expression between the groups in our present study (Table 6). Inducible nitric oxide synthase (*Nos2*) and NADPH oxidase 4 (*Nox4*) are major sources of elevated nitro-oxidative stress in the heart [34].

Here, *Nos2* was significantly overexpressed in all DOXO groups irrespective of treatments with losartan, mirabegron, and their combination (Table 6). There was no significant difference in the *Nox4* expression at the mRNA level between the DOXO-only and the control groups. Losartan did not affect the *Nox4* expression; however, mirabegron and the combination of losartan plus mirabegron markedly reduced the *Nox4* expression compared to those of the control or DOXO-only groups (Table 6). The left ventricular expressions of the free radical eliminating superoxide dismutase isoforms (*Sod1, Sod2*, and *Sod3*) were similar in the control, DOXO-only, and losartan-treated DOXO groups (Table 6). In response to mirabegron, the extracellular *Sod3* isoform was repressed compared to the control group (Table 6). In response to losartan plus mirabegron, all *Sod* isoforms were repressed compared to the control group (Table 6). The left ventricular expression of the hydrogen peroxide eliminating catalase (*Cat*) was also similar in the control, DOXO-only, and losartan-treated DOXO groups (Table 6). Notably, losartan treatment resulted in a trend toward decreasing *Cat* expression compared to the control group (Table 6). Moreover, the losartan plus mirabegron treatment significantly reduced the *Cat* expression compared to the values of the control or DOXO-only groups (Table 6).

### 2.10. Left Ventricular Overexpression of Inflammatory Markers Il1, Il6 and Tnf Were Ameliorated by Losartan and Mirabegron in DOXO-Induced Chronic Cardiotoxicity at Week 9

Inflammatory processes triggered by the over-activation of RAAS are reported to be major contributors to the development of cardiac remodeling and fibrosis in heart failure [35]. In our DOXO-induced chronic cardiotoxicity model, there was no significant difference in the left ventricular expression of the alternative angiotensin II activator chymase (*Cma*), angiotensinogen (*Agt*), and angiotensin II type 1 receptor (*Agtr1*) between the DOXO-only and control groups (Figure 7a–c). Notably, the *Agt* (*p* = 0.095) and *Agtr1* (*p* = 0.074) expressions showed a trend towards decreasing response to mirabegron and a significant reduction in response to losartan plus mirabegron compared to the control group (Figure 7a–c). The inflammatory markers interleukin-1 (*Il1*), interleukin-6 (*Il6*), and tumor necrosis factor-alpha (*Tnf*) were significantly overexpressed in the DOXO-only group compared to those of the control group (Figure 7d–f). Losartan significantly reduced the overexpression of *Il1* and *Il6* compared to the values of the DOXO-only group (Figure 7d–f). Mirabegron reduced the overexpression of all measured inflammatory markers compared to the DOXO-only group (Figure 7d–f).

In response to losartan plus mirabegron, *Il6* showed a decreasing tendency, whereas *Il1* and *Tnf* expressions were not significantly different from those of the DOXO-only group (Figure 7d–f).

## 3. Discussion

Here we report that mirabegron and its combination with losartan improved the systolic and diastolic dysfunction and reduced *Smad2* and *Smad3* overexpression in our DOXO-induced chronic cardiotoxicity model. Only mirabegron improved the DOXO-induced LV fibrosis markedly. Losartan failed to ameliorate the systolic dysfunction; however, it improved the diastolic dysfunction and prevented the SERCA2a repression in our DOXO-induced cardiotoxicity model. LV overexpression of *Il1b* and *Il6* was significantly reduced by losartan and mirabegron. Mirabegron and its combination with losartan seem to be promising therapeutic tools against systolic and diastolic dysfunction in DOXO-induced chronic cardiotoxicity.

Systolic and diastolic dysfunction resulting in heart failure are severe and well-known adverse effects of DOXO treatment [6,31]. In the present study, the mortality rate [36], echocardiographic parameters, heart rate [37,38], blood pressure [39,40], serum lipid [41,42], autopsy, and histologic findings in our DOXO-induced chronic cardiotoxicity model are consistent with the literature data on preclinical models [43] and similar to those seen in tumor survivor patients treated with DOXO [3,6,31]. Among the nine animals that died in the DOXO groups, five rats died before the start of the treatments or should be excluded due to poor systolic function (LVEF < 40%) assessed by echocardiography. During the treatments with losartan, mirabegron, and their combination, four animals died. These cases might be the consequences of DOXO treatment and not the side effects of the drugs administered in this study. To decide if the used drugs have severe side effects in DOXO-induced cardiotoxicity, more parameters, including arrhythmias, should be tested in more doses and follow-up time points with higher sample numbers in the groups.

In the cardio-oncology practice, secondary prevention of chemotherapy-induced chronic cardiotoxicity starts by using drugs applied in standard HF therapy when symptoms, increase in cardiac biomarkers, or systolic dysfunction develops [6,31]. In our present study, 4 weeks after the last DOXO cycle, early signs of systolic dysfunction (i.e., increased LV end-systolic diameter and decreased FS) were detected in the DOXO groups. Therefore, in our present study, treatments by losartan, mirabegron, and their combination started from week 5, mimicking the conventionally scheduled therapeutic regimens in chemotherapy-induced cardiotoxicity in tumor survivor patients. It should be mentioned that cancer patients at increased risk for chronic cardiotoxicity, defined by a history or risk factors of CVDs, previous cardiotoxicity, or exposure to cardiotoxic agents, may benefit from primary preventive HF strategies when the HF medication starts during or before the chemotherapy [6,31]. Our present study might better mimic the clinical situation when children or young adults without severe risk factors for CVDs are diagnosed with tumors and treated with DOXO. At the endpoint of our present study, the DOXO-induced LV wall thinning, systolic dysfunction, and cardiac fibrosis were associated with the overexpression of selected elements of the fibrotic TGF-β/SMAD signaling pathway (i.e., *Ctgf*, TGFβRII, *Smad2, Smad3,* and *Col1a1*) and molecular markers of apoptosis (i.e., *Bax*, and *Bax/Bcl2* ratio), cardiac remodeling (i.e., *Mmp2* and *Mmp9*), heart failure, (i.e., *Nppa* and *Nppb*), inflammation (i.e., *Il1, Il6,* and *Tnf*) and nitro-oxidative stress (i.e., *Nos2*). Moreover, the DOXO-induced diastolic dysfunction was accompanied by a reduced left ventricular SERCA2a level. These molecular findings are also in accordance with the literature data on DOXO-induced cardiotoxicity models [7,15,44,45,46,47]. Several DOXO-induced chronic cardiotoxicity models, particularly if using higher DOXO doses, may also develop severe kidney failure [43]. In our previous study, mirabegron significantly worsened the renal function in a rat model of chronic kidney disease [29]. Therefore, we aimed at using a DOXO-induced chronic cardiotoxicity model, which does not develop severe renal failure and consequently does not worsen the DOXO-induced heart failure (i.e., type 4 cardio-renal syndrome [48]). Indeed, the serum carbamide and creatinine levels were not significantly increased in the DOXO-treated groups compared to the control group in our present study.

According to the latest guideline of the European Society of Cardiology (ESC), RAAS inhibitors, including ARBs, are recommended for pharmacological therapy of HFrEF patients to reduce the risk of HF hospitalization and CV death [31]. Moreover, the latest ESC position paper on cancer treatments and cardiovascular toxicity stated that patients who develop asymptomatic LV dysfunction or HF during cancer therapy are likely to profit from the angiotensin-converting enzyme (ACE) inhibitors or ARBs and beta-blocker treatment similar to the general HF population [6]. Since our DOXO-induced cardiotoxicity model developed bradycardia, we avoided the administration of beta-blockers alone or in combination with ARBs. The ARB losartan showed antiremodeling and cardioprotective effects in our rat models of radiation-induced heart disease [49] and uremic cardiomyopathy [29], or DOXO-induced chronic cardiotoxicity models used by others [19,50,51]. In contrast, in our hands, losartan failed to significantly improve the morphologic parameters (i.e., systolic wall thicknesses, LV end-systolic diameter, cardiomyocyte cross-sectional area) and the systolic dysfunction (i.e., reduced FS and EF) in DOXO-induced chronic cardiotoxicity in the present study. A potential explanation for the lack of significant antiremodeling effects of losartan could be that losartan failed to significantly reduce the LV fibrosis and showed only a tendency to decrease in cardiac collagen content at the endpoint in our DOXO-induced cardiotoxicity model. Indeed, the LV expressions of selected elements of the TGF-β/SMAD fibrotic pathway (i.e., *Ctgf,* TGFβRII, *Smad3*, and *Col1a1*) were not significantly different in the losartan-treated DOXO group compared to the DOXO-only or control groups. In contrast to the results of Zong et al., using higher DOXO doses (ip. 6 × 2.5 mg/kg) and shorter follow-up time (i.e., 6 weeks) to induce chronic cardiotoxicity, *Agtr1a* failed to be overexpressed in the left ventricles of our DOXO-induced cardiotoxicity model [50]. This fact might provide another explanation for the lacking antiremodeling effects of the ARB losartan in our DOXO-induced cardiotoxicity model. Notably, losartan significantly shortened the heart rate-independent diastolic function parameter isovolumetric relaxation time (IVRT) in our DOXO-induced chronic cardiotoxicity model. SERCA was shown to determine the magnitude of myocyte Ca^2+^ cycling [52]. The early diastolic reuptake of Ca^2+^ into the sarcoplasmic reticulum, in part, determines the velocity at which the left ventricle relaxes (i.e., IVRT) [52]. Accordingly, losartan prevented the repression of SERCA2a associated with shorter IVRT in our DOXO-induced cardiotoxicity model. Moreover, it was reported that cytokines, particularly IL6, induced reciprocal expression of SERCA and natriuretic peptides at the mRNA level in cultured rat ventricular myocytes [53]. Indeed, in our present study, losartan significantly reduced DOXO-induced LV overexpression of inflammatory markers (i.e., *Il1* and *Il6*) and natriuretic peptides (i.e., *Nppa* and *Nppb*) in our DOXO-induced chronic cardiotoxicity model. In contrast, the mitral annulus velocity e’ and consequently the E/e’ ratio failed to be significantly improved by losartan in our DOXO model. Notably, cardiac fibrosis was shown to worsen myocardial relaxation parameters, including e’ and E/e’ [54]. Since losartan failed to significantly improve the LV fibrosis, and probably, as a consequence, the e’ and E/e’ were not different from those in the DOXO-only group.

Mirabegron is a β3AR agonist recently used in the treatment of overactive bladder syndrome in urology practice [21]. Interestingly, β3AR agonists showed beneficial effects on the symptoms of HF [21,55]. In healthy atrial and ventricular myocytes, the expression of β3AR is considered low but more abundant in non-cardiomyocytes, including endothelial cells [21,56,57]. In contrast to β1- and β2ARs, cardiac β3AR expression increases in chronic ischemia and heart failure, which is considered a counterregulatory mechanism to prevent chronic adrenergic overactivation [21,56,58,59,60]. Most studies investigating the effects of β3AR agonists in HF demonstrated that β3AR agonists attenuated cardiac fibrosis and improved cardiac contractility via the β3AR/eNOS/cGMP signaling pathway as the main mechanism [21,55]. In contrast, the left ventricular β3AR was significantly repressed in all DOXO groups independent of treatments, and the eNOS and p-eNOS levels and their ratios were similar in all groups in our present experiment. However, mirabegron significantly improved the morphologic (i.e., left ventricular wall thicknesses and end-systolic diameter), systolic (i.e., FS and EF), and several diastolic (i.e., e’ and E/e’) parameters in our DOXO-induced cardiotoxicity model. Importantly, in our present study, mirabegron was the only treatment that significantly reduced DOXO-induced cardiac fibrosis and the apoptosis marker *Bax/Bcl2* ratio. In our previous study in a rat model of uremic cardiomyopathy, mirabegron had a moderate antifibrotic effect associated with improved diastolic function independently of the β3AR/eNOS signaling pathway [29]. Indeed, β3AR agonists were shown to have beneficial effects independently of coupling the β3AR to eNOS in the heart and other tissues. These mechanisms include, e.g., antifibrotic effects via downregulation of the AT1 receptor [25,26,27,29], and CTGF [61], antioxidant and anti-inflammatory properties [62,63]. Therefore, we investigated the changes in the left ventricular expression of selected molecular markers of the RAAS, fibrosis, nitro-oxidative stress, and inflammation in response to losartan, mirabegron, and their combination in our DOXO-induced cardiotoxicity model. Interestingly, in response to mirabegron, the left ventricular expressions of the RAAS-associated *Agt and Agtr1a* failed to change, and the expressions of nitro-oxidative stress-associated *Nos2* and *Nox4*, and the fibrotic *Ctgf* and TGFβRII remained high in our DOXO-induced cardiotoxicity model. In contrast, mirabegron significantly decreased *Smad2* and *Smad3* expressions compared to the DOXO-only group, and the *Col1a1* expression was not markedly different from the control group. Therefore, we speculate here that mirabegron exerts its antiremodeling effect via inhibiting SMAD2/3-mediated fibrotic mechanisms independently of the β3AR/eNOS signaling pathway in our DOXO-induced chronic cardiotoxicity model.

To investigate if the beneficial effects of losartan and mirabegron are additive, the combination of losartan and mirabegron was administered in a group of DOXO-treated animals. In summary, the combination treatment preserved the systolic function (i.e., FS and EF) 8 weeks after the DOXO administration. However, it should be mentioned that both FS and EF tended to decrease, and the isovolumic contraction time showed an increasing tendency in the combination treatment group compared to the control group. However, the combination treatment improved the diastolic parameters e’ and E/e’, similarly to the effects of mirabegron alone in our DOXO-induced cardiotoxicity model. Notably, the combination treatment failed to prevent the repression of SERCA2a and significantly reduce the overexpression of the inflammatory markers (i.e., *Il1* and *Il6*) compared to the DOXO-only group. In response to the combination treatment, the cardiac collagen content and the expressions of *Col1a1, Ctgf* and *Smad2* were not significantly different compared to those of the control group, whereas *Smad3, Agtr1, Nox4, Sod2,* and *Nppb* expressions were significantly reduced compared to the DOXO-only group. The combination treatment significantly repressed the LV overexpression of the superoxide eliminating *Sod2* and *Sod3* and the hydrogen-peroxide eliminating *Cat*. This might be a consequence of the significant repression of the nitro-oxidative stress markers *Nox4* and *Sod2* compared to those of the DOXO-only group. Angiotensin-II is known to increase the nitro-oxidative stress and inflammation via AT1 (*Agtr1*) receptors by increasing NADPH-oxidase (*Nox4*) [64]. *Agtr1* was also significantly repressed in response to the combination treatment in our present study. Losartan and mirabegron seem to have a potentiating effect on reducing the AT1 receptor-mediated nitro-oxidative stress in DOXO-induced cardiotoxicity. However, this speculation should be proven by further experiments investigating the levels of ROS and RNS as well as RAAS and nitro-oxidative stress-associated proteins.

Our study is not without limitations. We intended to test and compare the effects of chronic administration of the ARB losartan, the β3AR agonist mirabegron, and their combination on cardiac remodeling in DOXO-induced chronic cardiotoxicity in rats. We used only single doses of losartan and mirabegron, which are comparable to the human therapeutic doses and used in heart failure models of other etiologies. We tested the effects of these aforementioned drugs only in secondary prevention. It could not be excluded that losartan might have beneficial effects if its administration starts in primary prevention (i.e., during or before the DOXO treatment). DOXO-induced chronic cardiotoxicity is a well-known complication most commonly manifested as left ventricular systolic dysfunction with reduced ejection fraction. Nevertheless, significant differences exist between the DOXO administration and pathomechanisms in the model used vs. that in patients. The time period between the consecutive DOXO cycles was shorter compared to the human protocols due to the lifespan of the rats. Although female gender is a risk factor in the development of DOXO-induced cardiotoxicity in humans [3,6,65], the standardization of the results independent of female sex hormonal effects is easier in males. In our present study, only healthy male rats without cancer, CVDs, or other comorbidities were used in order to investigate the isolated DOXO effects on the heart. In future studies, animal models of both sexes representing tumor development and receiving multimodality oncotherapy (i.e., radio- and chemotherapy) would be more suitable to mirror the clinical scenario of onco-cardiology patients as closely as possible. Moreover, there are still many unknown mechanisms, including the exact role of β3AR in the development of DOXO-induced chronic cardiotoxicity. We demonstrated here that LV β3AR expression decreased, and the *Agtr1* expression failed to change in DOXO-induced chronic cardiotoxicity 9 weeks after DOXO administration in our rat model. However, it could not be excluded that LV β3AR or *Agtr1* expressions might change at the protein level in earlier or later HF phases of DOXO-induced cardiotoxicity. The time-dependent and mechanistic investigation of the cardiac β3AR expression in the development of DOXO-induced chronic cardiotoxicity was out of the scope of the present study. Notably, the precise mechanisms and functional role of the β3AR in HF induced by different cardiovascular diseases, including diabetes mellitus, acute myocardial infarction, chronic kidney disease, or other chemotherapy-induced heart failure forms, are not fully discovered. We focused mainly on the potential protective effects of losartan, mirabegron, and their combination in DOXO-induced chronic cardiotoxicity, investigating the well-known markers of cardiac remodeling and the effects caused by losartan and mirabegron. The deep mechanistic insight of the cardioprotective effects caused by losartan and mirabegron was out of the scope of our present descriptive study. We found antiremodeling effects of mirabegron and hypothesized that mirabegron could have antifibrotic effects in DOXO-induced chronic cardiotoxicity independently of the β3AR-eNOS-mediated pathway, but probably associated with its effects causing repression on the elements of the fibrotic TGF-β/SMAD2/3 pathway. However, further mechanistic studies, including the inhibition of the TGF-β/SMAD2/3 pathway and investigation of the expression of phosphorylated SMAD2 and SMAD3, as well as endogenous negative regulators of this pathway such as SMAD6 and SMAD7, are needed to explore the antiremodeling effects of mirabegron in DOXO-induced chronic cardiotoxicity.

## 4. Materials and Methods

### 4.1. Ethics Approval

This investigation conformed to the EU Directive 2010/63/EU and was approved by the regional Animal Research Ethics Committee of Csongrád County (Csongrád county, Hungary; project license: XV./57/2020, date of approval: 12 February 2020). All institutional and national guidelines for the care and use of laboratory animals were followed.

### 4.2. Animals

A total of 50 male Wistar rats (350–400 g, 8–10 weeks old) were used in the experiments. The animals were housed in pairs in individually ventilated cages (Sealsafe IVC system, Buguggiate, Italy) in a temperature-controlled room (22 ± 2 °C; relative humidity 55 ± 10%) with a 12 h:12 h light/dark cycle. Standard rat chow and tap water were supplied ad libitum during the experiment.

### 4.3. Experimental Setup

After 1 week of acclimatization, the animals were randomly assigned to one control (*n* = 8) and four DOXO-treated groups (total *n* = 42, *n* = 10–11/group) (Figure 1). Control animals received saline (*ip.* 1 mL/kg), and rats in the DOXO group received *ip.* 1.5 mg/kg DOXO at days 1, 4, 7, 10, 13, and 16 (i.e., 9 mg/kg cumulative dose) (Figure 1). Timing [66,67,68] and dosing [36,69,70] of DOXO cycles were calculated based on preclinical protocols [43], and human chemotherapy regimens [71,72,73] corrected to the lifespan, body surface area, and metabolism of rats [74,75].

According to the conventional concept, heart failure therapy starts during or after chemotherapy if patients show early signs of LV systolic dysfunction assessed by echocardiography [6,31]. Therefore, in our present study, cardiac morphology and function were assessed by echocardiography at the fourth follow-up week to monitor if the early signs of DOXO-induced chronic cardiotoxicity developed (Figure 1). From the fifth follow-up week after the saline or DOXO treatments, the rats were treated via oral *gavage* daily until the end of the experiments at the 9th follow-up week as follows: (i) control group treated with tap water (*n* = 8; *per os* 2 mL/kg/day,); (ii) DOXO group treated with tap water (*n* = 11, *per os* 2 mL/kg/day,); (iii) DOXO group treated with losartan (*n* = 10; *per os* 20 mg/kg/day dissolved in tap water in 2 mL/kg end volume, Arbartan 50 mg film-coated tablets, Teva Pharmaceutical Industries Ltd., Debrecen, Hungary); (iv) DOXO group treated with mirabegron (*n* = 11, *per os* 30 mg/kg/day dissolved in tap water 2 mL/kg end volume; Betmiga 50 mg prolonged-release tablets, Astellas Pharma Europe B.V., Leiden, Netherland); and (v) DOXO group treated with the combination of losartan (*per os* 20 mg/kg/day dissolved in tap water in 2 mL/kg end volume, Arbartan 50 mg film-coated tablets, Teva Pharmaceutical Industries Ltd., Debrecen, Hungary) plus mirabegron (*per os* 30 mg/kg/day dissolved in tap water 2 mL/kg end volume; Betmiga 50 mg prolonged-release tablets, Astellas Pharma Europe B.V., Leiden, Netherland, *n* = 10, Figure 1). The doses of the drugs applied here were calculated from existing recommendations on losartan for human heart failure [31], and phase II clinical trials on mirabegron (trial numbers: NCT01876433 and NCT03926754) corrected to the metabolism and body surface area of rats [74]. At the eighth follow-up week, cardiac morphology and function were assessed by transthoracic echocardiography (Figure 1.) At the ninth follow-up week, rats were anesthetized with sodium pentobarbital (Euthasol, ip. 40 mg/kg Produlab Pharma b.v., Raamsdonksveer, The Netherlands), and invasive blood pressure measurement was performed (Figure 1). Then blood was collected from the abdominal aorta to measure routine laboratory parameters on cardiovascular risk factors and kidney function (Figure 1). After the blood sampling, hearts, lungs, and tibias were isolated, and the blood was washed out from the heart in calcium-free Krebs-Henseleit solution. Then left and right ventricles were separated, measured, and LV samples were prepared for histology and biochemical measurements. The development of chronic cardiotoxicity and LV fibrosis in the DOXO groups were checked by the measurement of cardiomyocyte cross-sectional areas on hematoxylin-eosin (HE)-stained slides and picrosirius red/fast green-stained (PSFG) slides (Figure 1). Total RNA was isolated from the left ventricles, and the expression of fibrosis (i.e., *Ctgf*, *Tgfb*, *Smad2, Smad3, Col1a1)*, cardiac remodeling (*Mmp2*, *Mmp9*), heart failure, (*Nppa,* and *Nppb*), RAAS-associated (i.e., *Cma*, *Agt,* and *Agtr1*), inflammatory (i.e., *Il1, Il6*, and *Tnf*) and nitro-oxidative stress markers (*Nos1, Nos2, Nos3, Sod1*, *Sod2, Sod3,* and *Cat*) were measured at the transcript level by qRT-PCR at week 9 (Figure 1). Moreover, left ventricular protein levels of SERCA2a, β3AR, eNOS, p-eNOS, and TGFβRII were measured by using Western blot technique at week 9.

### 4.4. Transthoracic Echocardiography

Cardiac morphology and function were assessed by transthoracic echocardiography as described previously [49] at weeks 4 and 8 to monitor the development of DOXO-induced chronic cardiotoxicity. Rats were anesthetized with 2% isoflurane (Forane, Aesica, Queenborough Limited, Queenborough, UK). Two-dimensional, M-mode, Doppler, tissue Doppler, and four chamber-view images were performed by the criteria of the American Society of Echocardiography with a Vivid IQ ultrasound system (General Electric Medical Systems, New York, NY, USA) using a phased array 5.0–11 MHz transducer (General Electric 12S-RS probe, General Electric Medical Systems, New York, NY, USA). Data of three consecutive heart cycles were analyzed (EchoPac Dimension v201, General Electric Medical Systems, USA; https://www.gehealthcare.com/products/ultrasound/vivid/echopac, accessed on 09 January 2022) by an experienced investigator in a blinded manner. The mean values of three measurements were calculated and used for statistical evaluation. Systolic and diastolic wall thicknesses were obtained from parasternal short-axis view at the level of the papillary muscles (in cases of anterior and inferior walls) and long-axis view at the level of the mitral valve (in cases of septal and posterior walls). Left ventricular diameters were measured by means of M-mode echocardiography from the long-axis view between the endocardial borders. The fractional shortening and ejection fraction were calculated on M-mode images in the long-axis view using the Teichholz method. Diastolic function was assessed using pulse-wave Doppler across the mitral valve and tissue Doppler on the septal mitral annulus from the apical four-chamber view. Early (E) flow and septal mitral annulus velocity (e’) indicate diastolic function. Isovolumetric relaxation time and isovolumic contraction time were measured in the pulse wave Doppler images.

### 4.5. Blood Pressure Measurement

To measure arterial blood pressure, a PE50 polyethylene catheter (Cole-Parmer, Vernon Hills, IL, USA) was inserted into the left femoral artery at week 9 under pentobarbital anesthesia (Euthasol, Produlab Pharma B.V., Raamsdonksveer, The Netherlands; 40 mg/kg) as described previously [29]. Blood pressure measurements were performed between 09:00 and 14:00 h.

### 4.6. Blood Serum Parameters

At week 9, serum carbamide and creatinine levels were measured to assess renal function. Serum carbamide and creatinine levels were quantified by kinetic UV method using urease and glutamate dehydrogenase enzymes and Jaffe method, respectively. All reagents and instruments for the serum parameter measurements were from Roche Diagnostics (Hoffmann-La Roche Ltd., Basel, Switzerland), as described previously [76]. Cardiovascular risk factors, including serum cholesterol and triglyceride levels, were also measured by Roche Cobas 8000 analyzer system using enzymatic colorimetric assays from Roche (Hoffmann-La Roche Ltd., Basel, Switzerland) [29].

### 4.7. Tissue Harvesting

At week 9, the hearts of the animals were isolated under pentobarbital anesthesia (Euthasol, ip. 40 mg/kg, Produlab Pharma b.v., Raamsdonksveer, The Netherlands), and the blood was washed out in calcium-free Krebs-Henseleit solution. Then the hearts were weighed, left and right ventricles were separated and weighed, and the cross-section of the left ventricle at the ring of the papillae was cut and fixed in 4% buffered formalin for histological analysis. Other parts of the left ventricles were freshly frozen in liquid nitrogen and stored at −80 °C until further biochemical measurements. Body weight, tibia length, and weights of the lungs were also measured.

### 4.8. Hematoxylin-Eosin and Picrosirius Red and Fast Green Stainings

Five μm thick transverse cut sections of the formalin-fixed and paraffin-embedded subvalvular left ventricular areas were stained with hematoxylin-eosin (HE) or picrosirius red and fast green (PSFG) as described previously [77]. Histological slides were scanned with a Pannoramic Midi II scanner (3D-Histech, Budapest, Hungary). Representative HE- and PSFG-stained slides were captured in Panoramic Viewer 1.15.4 (3D-Histech, Budapest, Hungary; https://old.3dhistech.com/pannoramic_viewer, accessed on 9 January 2022).

On the digital HE images, cardiomyocyte cross-sectional areas were measured to assess the ventricular wall thinning at the cellular level. For the evaluation, Biology Image Analysis Software (BIAS) was used [29,49]. BIAS (internal built, dated December 2019; https://single-cell-technologies.com/bias/, accessed on 9 January 2022) is developed by Single-Cell Technologies Ltd., Szeged, Hungary, and the first publicly available version is expected to be released in early 2022. Image preprocessing was followed by deep learning-based cytoplasm segmentation. User-selected objects were forwarded to the feature extraction module, configurable to extract properties from the selected cell components. Transverse transnuclear cardiomyocyte perimeters were measured in 100 (consecutive) cardiomyocytes selected on the basis of longitudinal orientation and mononucleation from a single cut surface (digitalized histological slide) of the left ventricular tissue blocks.

Cardiac fibrosis was assessed on PSFG slides with an in-house developed program [78,79]. This program determines the proportion of red pixels of heart sections using two simple color filters. For each red–green–blue (RGB) pixel, the program calculates the color of the pixel in hue–saturation–luminance (HSL) color space. The first filter is used for detecting red portions of the image. The second filter excludes any white (empty) or light grey (residual dirt on the slide) pixels from further processing using a simple RGB threshold. In this way, the program groups each pixel into one of two sets: pixels considered red, and pixels considered green but neither white nor grey. Red pixels in the first set represent collagen content and fibrosis. Green pixels in the second set correspond to cardiac muscle. The mean values of 10 representative images were calculated and used for statistical evaluation in the case of each left ventricular slide. Medium-size vessels and their perivascular connective tissue sheet, the subepicardial and subendocardial areas were avoided as much as possible.

### 4.9. mRNA Expression Profiling by qRT-PCR

Quantitative RT-PCR was performed with gene-specific primers to monitor mRNA expression as described previously [77]. Briefly, RNA was isolated using Qiagen RNeasy Fibrous Tissue Mini Kit (Qiagen, Hilden, Germany) from heart tissue. Then, 100 μg of total RNA was reverse transcribed using iScript™ cDNA Synthesis Kit (BioRad Laboratories Inc., Hercules, CA, USA). Specific primers (*Agt*: angiotensinogen, #qRnoCED0051666; *Agtr1*: angiotensin II receptor type 1, #qRnoCID0052626; *Bax*: BCL2-associated X apoptosis regulator, #qRnoCED0002625; *Bcl2:* B-Cell CLL/lymphoma 2 apopotosis regulator, #qRnoCED0006419; *Cat*: catalase, #qRnoCID0006360; *Cma1*: chymase, #qRnoCED0005462; *Col1a1*: collagen type 1 alpha 1 chain, #qRnoCED0007857; *Ctgf*: connective tissue growth factor, #qRnoCED0001593; *Il1*: interleukin-1, #qRnoCID0002056; *Il6*: interleukin-6, #qRnoCID0053166; *Mmp2:* matrix metalloproteinase 2, #qRnoCID0002887; *Mmp9:* matrix metalloproteinase 9, #qRnoCED0001183; *Nos1*: neuronal nitric oxide synthase, #qRnoCED0009301; *Nos2*: inducible nitric oxide synthase, #qRnoCID0017722; *Nos3*: endothelial nitric oxide synthase, #qRnoCID0005021; *Nox4:* NADPH-oxidase type 4, #qRnoCID0003969; *Nppa*: A-type natriuretic peptide, #qRnoCED0006216, *Nppb*: B-type natriuretic peptide, #qRnoCED0001541; *Smad2*, mothers against decapentaplegic homolog 2, #qRnoCID0005549; *Smad3*: mothers against decapentaplegic homolog 3, #qRnoCID0004164; *Sod1*: Cu/Zn superoxide dismutase (soluble) #qRnoCID0051055; *Sod2*: Mn superoxide dismutase (mitochondrial), #qRnoCID0008099; Sod3: Cu/Zn superoxide dismutase (extracellular), #qRnoCID0006360; *Tgfb*: transforming growth factor-β, #qRnoCID0009191, *Tnf*: tumor necrosis factor-α, #qRnoCED0009117) and SsoAdvanced™ Universal SYBR^®^ Green Supermix (BioRad Laboratories Inc., Hercules, CA, USA) were used according to the manufacturer’s instructions. Peptidyl-prolyl isomerase A (*Ppia*, forward primer sequence: *tgctggaccaaacacaaatg* and reverse primer sequence: *caccttcccaaagaccacat*) was used as a housekeeping control gene for normalization.

### 4.10. Western Blot

To investigate gene expression changes at the protein level, a standard Western blot technique was used as described previously [29,49]. TGFβRII (85 kDa) with GAPDH (37 kDa) loading background and SERCA2a (114 kDa), β3AR (44 kDa), eNOS (140 kDa), and p-eNOS (140 kDa) with α-tubulin (52 kDa) loading background were assessed at week 9. Left ventricular samples (*n* = 5–6 in each group, total *n* = 28) were homogenized with an ultrasonicator (UP100H, Hielscher, Germany) in Radio-Immunoprecipitation Assay (RIPA) buffer (50 mM Tris-HCl (pH 8.0), 150 mM NaCl, 0.5% sodium deoxycholate, 5 mM ethylenediamine tetra-acetic acid (EDTA), 0.1% sodium dodecyl sulfate, 1% NP-40; Cell Signaling Technology Inc., Danvers, MA, USA) supplemented with phenylmethanesulfonyl fluoride (PMSF; Sigma-Aldrich, St. Louis, MO, USA) and sodium fluoride (NaF; Sigma-Aldrich, Saint Louis, MO, USA). The crude homogenates were centrifuged at 15,000 × *g* for 30 min at 4 °C. After quantifying the supernatants’ protein concentrations using the BCA Protein Assay Kit (Pierce Thermo Fisher Scientific Inc., Waltham, MA, USA), 25 μg of reduced and denaturized protein was loaded. Then, sodium dodecyl-sulfate polyacrylamide gel electrophoresis (SDS-PAGE, 50 V, 4 h) was performed (6% gel in case of eNOS, p-eNOS, and SERCA2a, 10% gel in case of β3AR, and 12% gel in case of TGFβRII) followed by the transfer of proteins onto a nitrocellulose membrane (10% methanol in case of eNOS, p-eNOS, SERCA2a, and TGFβRII and 20% methanol in case of β3AR, 35 V, 2 h). The efficacy of transfer was checked using Ponceau staining. The membranes were cut horizontally into parts corresponding to the molecular weights of eNOS, p-eNOS, SERCA, β3AR, TGFβRII, GAPDH, and α-tubulin. Membranes were blocked for 1 h in 5% (*w/v*) bovine serum albumin (BSA, Sigma-Aldrich, St. Louis, MO, USA) and were incubated with primary antibodies in the concentrations of 1:1000 against eNOS (#32027S, Cell Signaling Technology Inc., Danvers, MA, USA), SERCA2a (#4388S, Cell Signaling Technology Inc., Danvers, MA, USA), TGFβRII (#79424T, Cell Signaling Technology Inc., Danvers, MA, USA) and α-tubulin (#2144S, Cell Signaling Technology Inc., Danvers, MA, USA) or 1:5000 against GAPDH (#2118, Cell Signaling Technology Inc., Danvers, MA, USA) or 1:500 against p-eNOS (Ser1177, #9570S, Cell Signaling Technology Inc., Danvers, MA, USA) and β3AR (AB101095, Abcam PLC, Cambridge, UK) overnight at 4 °C in 5% BSA. Then, the membranes were incubated with IRDye^®^ 800CW Goat Anti-Rabbit secondary antibody (LI-COR Biosciences, Lincoln, NE, USA, in the concentrations of 1:20,000 of TGFβRII and 1:5000 in the cases of the other investigated proteins) for 1 h at room temperature in 5% BSA antibodies to detect proteins. Fluorescent signals were detected by Odyssey CLx machine (LI-COR Biosciences, Lincoln, NE, USA), and digital images were analyzed and evaluated by Quantity One Software (Bio-Rad Laboratories Inc., Hercules, CA, USA). The full-length Ponceau-stained membranes and Western blots are presented in Appendix A. (The Appendix A also contains a more detailed description of our WB method from the steps of Ponceau staining to the fluorescent signal detection).

### 4.11. Statistical Analysis

Statistical analysis was performed using Sigmaplot 14.0 for Windows (Systat Software Inc., San Jose, CA, USA). All values are presented as mean ± SEM. *p* < 0.05 was accepted as a statistically significant difference. The corresponding table or figure legend describes specific sample numbers and statistical tests used for measurements. The normal distribution of the data was checked by the Shapiro-Wilk normality test. In the case of normal distribution, one-way ANOVA was used to determine the statistical significance between the measured parameters. If the normality test failed, the statistical program started Kruskal–Wallis test by ranks (i.e., ANOVA on ranks) automatically. In cases of significant differences between the groups, the Holm-Sidak test was used as *post hoc* test.

## 5. Conclusions

In this study, we evaluated the effects of chronic administration of the selective AT1 receptor blocker losartan, the β3AR agonist mirabegron, and their combination on LV morphology, function, and molecular markers of cardiac fibrosis, remodeling, heart failure, inflammation, and nitro-oxidative stress in a rat model of DOXO-induced chronic cardiotoxicity (Figure 8). Our results suggest that the development of DOXO-induced systolic and diastolic dysfunction may be prevented or markedly slowed down by mirabegron and the combination of mirabegron plus losartan if their administration started in the early stages of DOXO-induced chronic cardiotoxicity without a severe decrease in LVEF. Notably, the results on the ejection fraction, fractional shortening, left ventricular end-systolic diameters, several systolic wall thicknesses, and cardiac collagen content of the losartan plus mirabegron-treated group were not significantly different from those of either the control group or the DOXO-only group. However, further experiments with longer follow-up time, more doses, and higher sample numbers are needed to decide if the combination treatment could significantly ameliorate the systolic dysfunction in DOXO-induced chronic cardiotoxicity. Only mirabegron prevented the development of DOXO-induced LV fibrosis in our model. These antiremodeling effects of mirabegron seem to be independent of the β3AR/eNOS mediated pathways in DOXO-induced chronic cardiotoxicity, and probably associated with the repression on the SMAD2/3-mediated fibrotic pathway. In contrast, losartan failed to improve the severity of systolic dysfunction but had beneficial effects on diastolic dysfunction, SERCA2a level, and tissue inflammation in our DOXO-induced chronic cardiotoxicity model. In order to clarify whether mirabegron and the combination of mirabegron plus losartan (or other ARBs) may be protective against DOXO-induced chronic cardiotoxicity in humans, clinical trials enrolling a large number of patients are required for widening the indication of these otherwise routinely used drugs of heart failure treatment.

## Figures and Tables

**Figure 1 ijms-23-02201-f001:**
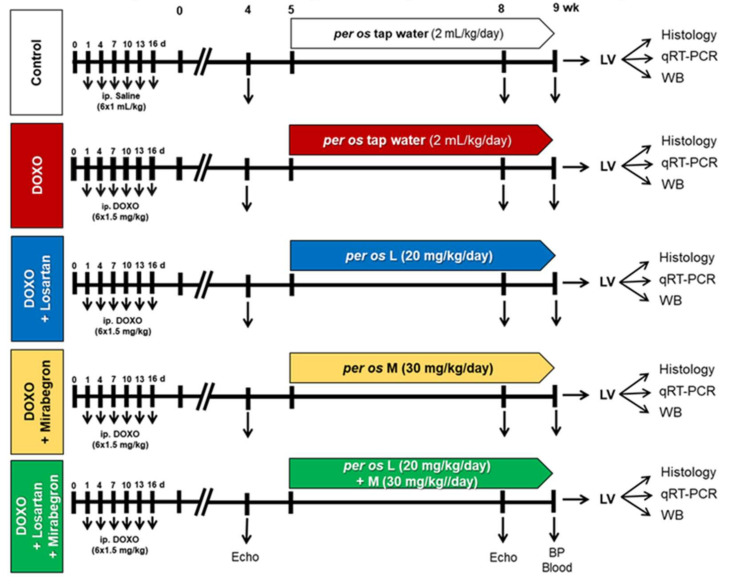
Experimental protocol. Male Wistar rats (*n* = 50, 350–400 g) were divided into one control and four doxorubicin (DOXO)-treated groups (*ip.* 1.5 mg/kg in 6 cycles; cumulative dose: 9 mg/kg). From the 5th week after the last cycle of DOXO administration, rats were treated via oral *gavage* daily until the end of the experiments as follows: (i) control group treated with tap water (*n* = 8, 2 mL/kg/day), (ii) DOXO-only group treated with tap water (*n* = 11, 2 mL/kg/day), (iii) DOXO group treated with losartan (L, *per os* 20 mg/kg/day, *n* = 10) dissolved in tap water, iv) DOXO group treated with mirabegron (M, *per os* 30 mg/kg/day, *n* = 10) dissolved in tap water, and (v) DOXO group treated with the combination of losartan (*per os* 20 mg/kg/day) and mirabegron (*per os* 30 mg/kg/day) dissolved in tap water (*n* = 11, 2 mL/kg/day). Cardiac morphology and function were assessed by transthoracic echocardiography (Echo) at weeks 4 and 8 under isoflurane anesthesia. At week 9, an invasive blood pressure (BP) measurement was performed under pentobarbital anesthesia, then blood was collected from the abdominal aorta to measure routine laboratory parameters, and hearts, lungs, and tibias were isolated. Left and right ventricles were separated, and left ventricular samples were prepared for histology, qRT-PCR, and Western blot (WB) measurements.

**Figure 2 ijms-23-02201-f002:**
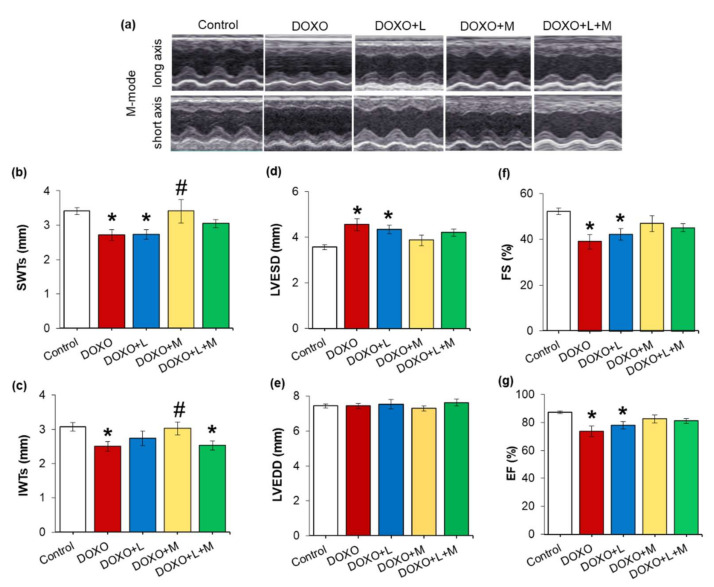
Effects of losartan, mirabegron, and their combination on morphological changes and systolic dysfunction in DOXO-induced chronic cardiotoxicity assessed by echocardiography at week 8. Values are presented as mean ± S.E.M., * *p*  <  0.05 vs. control group, # *p*  <  0.05 vs. DOXO-only group (*n* = 6–9, one-way ANOVA, Holm-Sidak *post hoc* test). (**a**) Representative M-mode images, (**b**) systolic septal wall thickness (SWTs), (**c**) systolic inferior wall thickness (IWTs), (**d**) left ventricular end-systolic diameter (LVESD), (**e**) left ventricular end-diastolic diameter (LVEDD), (**f**) fractional shortening (FS), and (**g**) ejection fraction (EF). DOXO: doxorubicin, L: losartan, M: mirabegron.

**Figure 3 ijms-23-02201-f003:**
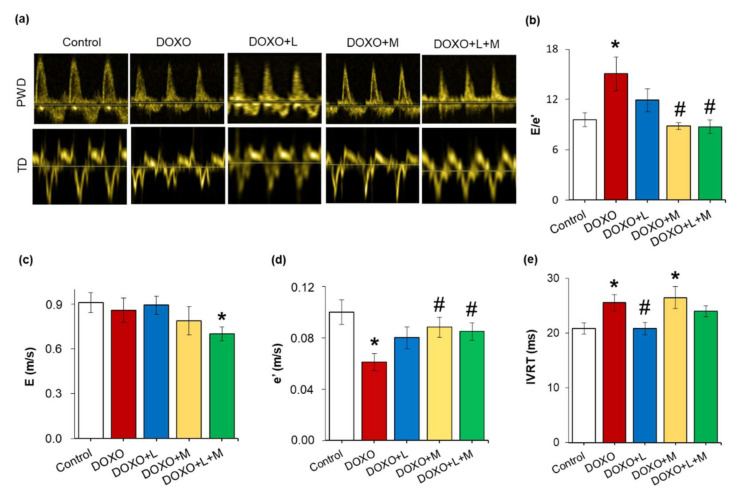
Effects of losartan, mirabegron, and their combination on diastolic dysfunction in DOXO-induced chronic cardiotoxicity assessed by echocardiography at week 8. Values are presented as mean ± S.E.M., * *p*  <  0.05 vs. control group, # *p*  <  0.05 vs. DOXO-only group (*n* = 6–9, one-way ANOVA or ANOVA on ranks in the case of E/e’, Holm-Sidak *post hoc* test). (**a**) representative pulse wave Doppler (PWD) and tissue Doppler (TD) images, (**b**) mitral valve early flow velocity (E) to septal mitral annulus velocity (e’) ratio (E/e’), (**c**) mitral valve early flow velocity (E), (**d**) septal mitral annulus velocity (e’), and (**e**) isovolumic relaxation time (IVRT). DOXO: doxorubicin, L: losartan, M: mirabegron.

**Figure 4 ijms-23-02201-f004:**
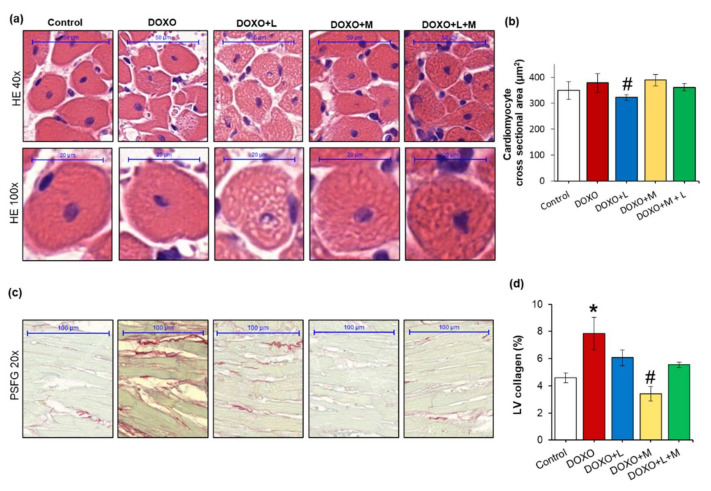
Effects of losartan, mirabegron, and their combination on DOXO-induced histologic changes at week 9. Values are presented as mean ± S.E.M., * *p*  <  0.05 vs. control group, # *p*  <  0.05 vs. DOXO-only group (*n* = 6–9, one-way ANOVA in the case of cardiomyocyte cross-sectional area and ANOVA on ranks in the case of LV collagen, Holm-Sidak post hoc test). (**a**) Representative hematoxylin-eosin (HE)-stained sections at 40× and 100× magnifications, (**b**) cardiomyocyte cross-sectional area, (**c**) representative picrosirius red and fast green (PSFG)-stained sections at 20× magnification, (**d**) left ventricular (LV) collagen content. DOXO: doxorubicin, L: losartan, M: mirabegron.

**Figure 5 ijms-23-02201-f005:**
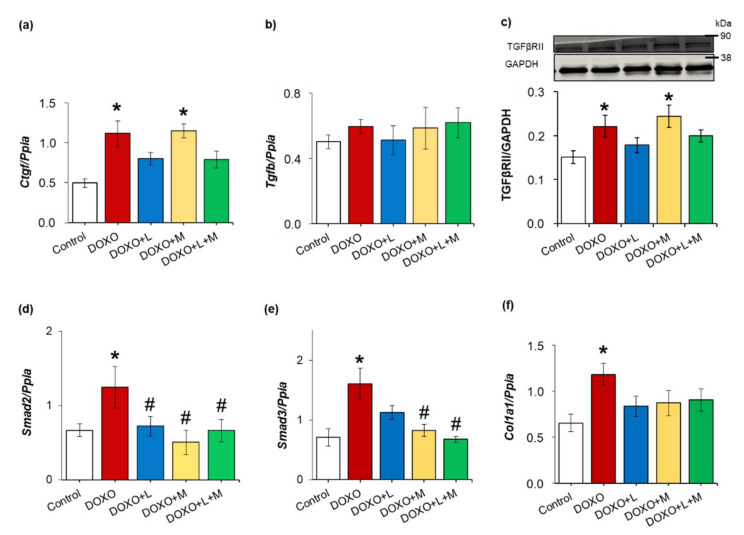
The effects of losartan, mirabegron, and their combination on the left ventricular expression of several elements of the TGF-β/SMAD fibrotic pathway in our DOXO-induced chronic cardiotoxicity model at week 9. (**a**) Left ventricular mRNA expressions of connective tissue growth factor (*Ctgf*) and (**b**) transforming growth factor-beta (*Tgfb*); (**c**) left ventricular protein expression and cropped representative Western blot images of (**c**) transforming growth factor-beta receptor type II (TGFβRII); (**d**) left ventricular mRNA expressions of mothers against decapentaplegic homolog 2 (*Smad2*); (**e**) mothers against decapentaplegic homolog 3 (*Smad3*), and (**f**) collagen 1a1 (*Col1a1*). Values are presented as mean ± S.E.M., * *p*  <  0.05 vs. control group, # *p*  <  0.05 vs. DOXO-only group (*n* = 5–9 for qPCR and *n* = 5–6 for Western blot results, ANOVA on ranks in the cases of *Tgfb* and TGFβRII expressions, and one-way ANOVA for the other parameters, Holm-Sidak post hoc test). DOXO: doxorubicin, GAPDH: glyceraldehyde 3-phosphate dehydrogenase (loading control in Western blot measurements), L: losartan, M: mirabegron. *Ppia*: peptidylprolyl isomerase A was used as the housekeeping gene for normalization in qPCR measurements. Western blot images were captured with the Odyssey CLx machine and exported with Image Studio 5.2.5 software. The full-length Ponceau-stained membranes and Western blots are presented in Appendix A.

**Figure 6 ijms-23-02201-f006:**
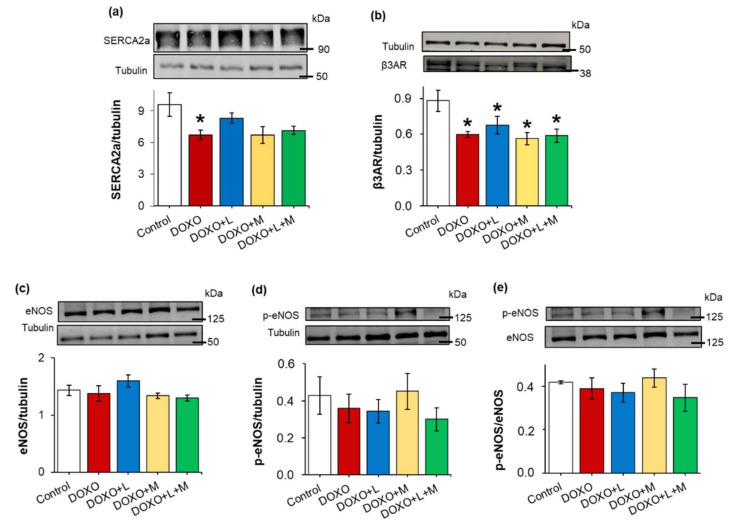
Effects of losartan, mirabegron, or their combination on the left ventricular expression of SERCA2a, β3AR, eNOS and p-eNOS in DOXO-induced cardiotoxicity at week 9. Left ventricular expression and cropped representative images of (**a**) sarcoendoplasmic reticulum calcium ATPase 2a (SERCA2a); (**b**) beta-3 adrenoceptor (β3AR); (**c**) endothelial nitric oxide synthase (eNOS); (**d**) phospho-eNOS (p-eNOS), and (**e**) p-eNOS/eNOS ratio. Values are presented as mean ± S.E.M., * *p*  <  0.05 vs. control group (*n* = 5–6, ANOVA on ranks in the cases of SERCA2a and p-eNOS expressions, and one-way ANOVA for the other parameters, Holm-Sidak post hoc test). Tubulin was used as a loading control. DOXO: doxorubicin, L: losartan, M: mirabegron. Images were captured with the Odyssey CLx machine and exported with Image Studio 5.2.5 software. The full-length Ponceau-stained membranes and Western blots are presented in Appendix A.

**Figure 7 ijms-23-02201-f007:**
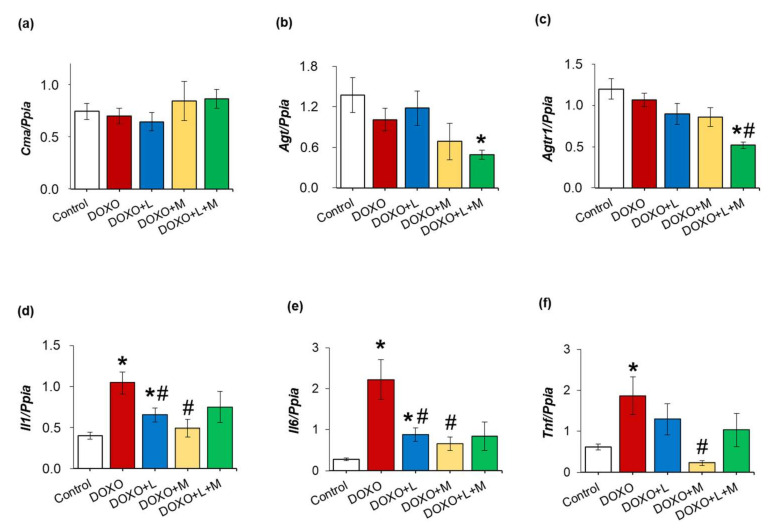
Effects of losartan, mirabegron, or their combination on the left ventricular expression of selected elements of the tissue renin-angiotensin-aldosterone system and inflammatory markers in DOXO-induced cardiotoxicity at week 9. Left ventricular expressions of (**a**) chymase (*Cma*); (**b**) angiotensinogen (*Agt*); (**c**) angiotensin II receptor type 1 (*Agtr1*); (**d**) interleukin-1 (*Il1*); (**e**) interleukin-6 (*Il6*), and (**f**) tumor necrosis factor-alpha (*Tnf*). Values are presented as mean ± S.E.M., * *p*  <  0.05 vs. control group, # *p*  <  0.05 vs. DOXO-only group (*n* = 5–6, ANOVA on ranks in the cases of Il6 and Tnf expressions, and one-way ANOVA for the other parameters, Holm-Sidak post hoc test). Peptidylprolyl isomerase A (*Ppia*) was used as the housekeeping gene for normalization DOXO: doxorubicin, L: losartan, M: mirabegron.

**Figure 8 ijms-23-02201-f008:**
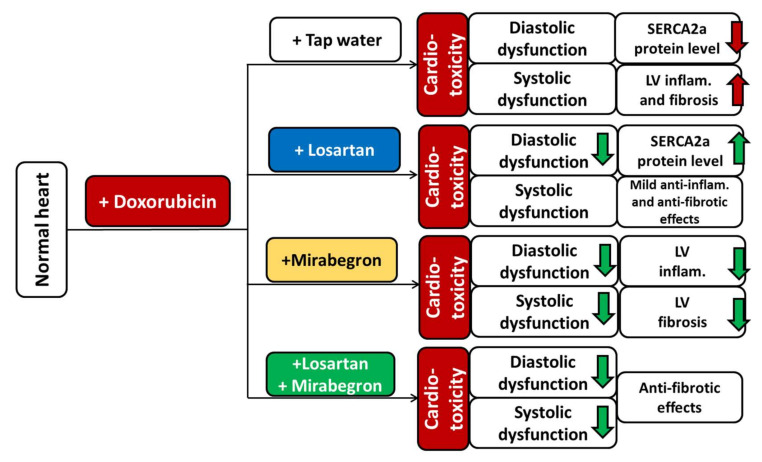
Summary figure; please see the conclusions for further explanations. Inflam: inflammatory, LV: left ventricular, SERCA2a: sarcoendoplasmic reticulum calcium ATPase.

**Table 1 ijms-23-02201-t001:** Effects of DOXO on cardiac morphology and function assessed by transthoracic echocardiography before starting the treatments with losartan, mirabegron, and their combination at week 4.

Parameter (Unit)	Groups				
	Control	DOXO	DOXO + L	DOXO + M	DOXO + L + M
SWTs (mm)	3.57 ± 0.06	3.43 ± 0.09	3.35 ± 0.19	3.53 ± 0.15	3.34 ± 0.16
SWTd (mm)	2.17 ± 0.04	1.92 ± 0.09	1.90 ± 0.09	1.99 ± 0.09	1.98 ± 0.14
PWTs (mm)	3.1 ± 0.11	2.82 ± 0.10	2.97 ± 0.15	3.16 ± 0.12	3.09 ± 0.10
PWTd (mm)	1.76 ± 0.05	1.73 ± 0.09	1.86 ± 0.07	1.91 ± 0.05	1.83 ± 0.02
AWTs (mm)	3.34 ± 0.11	3.05 ± 0.13	3.09 ± 0.15	3.21 ± 0.16	3.04 ± 0.12
AWTd (mm)	2.01 ± 0.08	1.84 ± 0.08	1.84 ± 0.09	1.88 ± 0.07	1.82 ± 0.09
IWTs (mm)	3.09 ± 0.1	2.88 ± 0.17	2.93 ± 0.11	3.01 ± 0.23	2.97 ± 0.07
IWTd (mm)	1.81 ± 0.07	1.85 ± 0.15	1.81 ± 0.1	1.94 ± 0.13	1.86 ± 0.08
LVEDD (mm)	7.02 ± 0.26	7.07 ± 0.21	7.10 ± 0.19	6.72 ± 0.24	6.86 ± 0.31
LVESD (mm)	3.04 ± 0.17	3.55 ± 0.23 *	3.62 ± 0.25 *	3.41 ± 0.08 *	3.43 ± 0.31 *
FS (%)	57 ± 1	50 ± 2	49 ± 3	53 ± 2	51 ± 3
EF (%)	90 ± 1	85 ± 1	84 ± 3	88 ± 1	86 ± 2
E (m/s)	0.90 ± 0.04	0.92 ± 0.06	0.93 ± 0.02	0.91 ± 0.08	0.92 ± 0.05
e’ (m/s)	0.089 ± 0.009	0.076 ± 0.008	0.066 ± 0.007	0.064 ± 0.009	0.085 ± 0.014
E/e’	10 ± 2	13 ± 1	15 ± 2	15 ± 3	13 ± 3
IVRT (ms)	17 ± 0.64	18 ± 0.81	16 ± 0.65	16 ± 0.95	17 ± 0.7
IVCT (ms)	16 ± 0.67	15 ± 0.67	15 ± 0.47	16 ± 0.99	15 ± 0.78
HR (1/min)	365 ± 9	368 ± 9	363 ± 11	382 ± 15	381 ± 7

Values are presented as mean ± S.E.M., * *p* < 0.05 vs. control group (*n* = 8–9, ANOVA on ranks in cases of IWTd, PWTd, E, e’, and E/e’, and one-way ANOVA for the other parameters, Holm-Sidak *post hoc* test). AWT: anterior wall thickness, d: diastole, DOXO: doxorubicin, E-velocity: early ventricular filling velocity, e’-velocity: diastolic septal mitral annulus velocity, EF: ejection fraction, FS: fractional shortening, HR: heart rate, IVCT: isovolumic relaxation time, IVRT: isovolumic contraction time, IWT: inferior wall thickness, L: losartan, LVEDD: left ventricular end-diastolic diameter, LVESD: left ventricular end-systolic diameter, M: mirabegron, PWT: posterior wall thickness, s: systole, SWT: septal wall thickness.

**Table 2 ijms-23-02201-t002:** Effects of losartan, mirabegron, and their combination on routine laboratory and clinical parameters in our DOXO-induced chronic cardiotoxicity model.

Parameter (Unit)	Groups				
	Control	DOXO	DOXO + L	DOXO + M	DOXO + L + M
Body weight before DOXO (g)	362 ± 10	357 ± 8	360 ± 11	364 ± 9	358 ± 9
Body weight after DOXO at week 0 (g)	394 ± 8	353 ± 11 *	355 ± 9 *	367 ± 9 *	363 ± 9 *
Body weight at week 9 (g)	451 ± 7	372 ± 19 *	360 ± 13 *	351 ± 18 *	379 ± 11 *
Serum carbamide (mmol/L)	7.76 ± 0.53	9.07 ± 0.80	7.93 ± 0.74	9.66 ± 1.82	9.23 ± 1.03
Serum creatinine (μmol/L)	36 ± 1.91	36 ± 3.59	35 ± 2.17	32 ± 3.44	27 ± 1.96
Serum cholesterol (mmol/L)	1.75 ± 0.07	8.91 ± 1.08 *	6.14 ± 0.93 *	6.01 ± 1.07 *	6.96 ± 1.20 *
Serum triglyceride (mmol/L)	0.69 ± 0.06	2.94 ± 0.30 *	2.71 ± 0.50 *	1.19 ± 0.30 ^#^	1.79 ± 0.24 *
SBP (mmHg)	148 ± 5	146 ± 8	119 ± 5 *^#^	153 ± 8	128 ± 6
DBP (mmHg)	108 ± 4	111 ± 5	79 ± 5 *^#^	116 ± 8	92 ± 4
MBP (mmHg)	122 ± 4	123 ± 7	98 ± 4 *^#^	131 ± 8	106 ± 5

Values are presented as mean ± S.E.M., * *p*  <  0.05 vs. control group, # *p*  <  0.05 vs. DOXO-only group (*n* = 10–11 for body weight before and after DOXO administration, *n* = 6–9 for body weight and serum parameters at week 9, and *n* = 5–8 for blood pressure parameters at week 9, ANOVA on ranks in cases of serum carbamide and creatinine levels, and one way ANOVA for the other parameters, Holm-Sidak *post hoc* test). DOXO: doxorubicin, DBP: diastolic blood pressure, L: losartan, M: mirabegron, MBP: mean arterial blood pressure, SBP: systolic blood pressure.

**Table 3 ijms-23-02201-t003:** Effects of losartan, mirabegron, or their combination on DOXO-induced cardiac morphologic and functional changes assessed by echocardiography at week 8.

Parameter (Unit)	Groups				
	Control	DOXO	DOXO + L	DOXO + M	DOXO + L + M
SWTd (mm)	2.10 ± 0.04	1.69 ± 0.09 *	1.76 ± 0.1	2.32 ± 0.25 ^#^	1.83 ± 0.1
PWTs (mm)	3.10 ± 0.09	2.52 ± 0.13 *	2.79 ± 0.18	3.01 ± 0.22	2.72 ± 0.18
PWTd (mm)	1.79 ± 0.12	1.69 ± 0.11	1.90 ± 0.10	1.92 ± 0.12	1.67 ± 0.13
AWTs (mm)	3.04 ± 0.10	2.57 ± 0.12 *	2.61 ± 0.19 *	3.21 ± 0.17 ^#^	3.31 ± 0.22 ^#^
AWTd (mm)	1.71 ± 0.07	1.60 ± 0.06	1.63 ± 0.16	2.03 ± 0.16 ^#^	2.02 ± 0.14 ^#^
IWTd (mm)	1.75 ± 0.07	1.57 ± 0.13	1.65 ± 0.1	1.84 ± 0.16	1.64 ± 0.1
IVCT (ms)	16 ± 2	21 ± 1 *	16 ± 1 ^#^	18 ± 2	20 ± 1
HR (1/min)	371 ± 11	347 ± 12	349 ± 15	343 ± 12	335 ± 10 ^#^

Values are presented as mean ± S.E.M., * *p*  <  0.05 vs. control group, # *p*  <  0.05 vs. DOXO-only group (*n* = 8–9, ANOVA on ranks in cases of PWTd and IVCT, and one-way ANOVA for the other parameters, Holm-Sidak *post hoc* test). AWT: anterior wall thickness, d: diastole, DOXO: doxorubicin, HR: heart rate, IVCT: isovolumic contraction time, IWT: inferior wall thickness, L: losartan, M: mirabegron, PWT: posterior wall thickness, s: systole, SWT: septal wall thickness.

**Table 4 ijms-23-02201-t004:** Effects of losartan, mirabegron, and their combination on tibia length, heart, and lung weights in our DOXO-induced chronic cardiotoxicity model at week 9.

Parameter (Unit)	Groups				
	Control	DOXO	DOXO + L	DOXO + M	DOXO + L + M
Tibia length (cm)	4.13 ± 0.02	4.06 ± 0.03	4.04 ± 0.04	4.07 ± 0.05	4.09 ± 0.03
Heart weight (mg)	1167 ± 20	972 ± 43 *	989 ± 52 *	1076 ± 75	1011 ± 27 *
LV weight (mg)	838 ± 15	693 ± 31 *	690 ± 36 *	768 ± 55	698 ± 22 *
RV weight (mg)	218 ± 12	178 ± 12 *	190 ± 17	181 ± 14	184 ± 4 *
Lung weight (mg)	1581 ± 29	1497 ± 44	1615 ± 60	1665 ± 106	1684 ± 52

Values are presented as mean ± S.E.M., * *p*  <  0.05 vs. control group (*n* = 6–9, ANOVA on ranks in the cases of heart and RV weights, and one-way ANOVA for the other parameters, Holm-Sidak *post hoc* test). L: losartan, LV: left ventricle, M: mirabegron, RV: right ventricle.

**Table 5 ijms-23-02201-t005:** The effects of losartan, mirabegron, and their combination on left ventricular expression of matrix metalloprotease-2 and -9, natriuretic peptides A and B, and apoptotic markers at the mRNA level in our DOXO-induced chronic cardiotoxicity model at week 9.

Relative Gene Expression	Groups				
	Control	DOXO	DOXO + L	DOXO + M	DOXO + L + M
*Mmp2/Ppia*	1.04 ± 0.07	1.71 ± 0.22 *	1.19 ± 0.15	1.29 ± 0.18	1.12 ± 0.28
*Mmp9/Ppia*	0.75 ± 0.06	1.5 ± 0.17 *	1.55 ± 0.28 *	1.27 ± 0.22	0.92 ± 0.08 #
*Nppa/Ppia*	0.26 ± 0.05	1.49 ± 0.28 *	0.78 ± 0.15 #	1.59 ± 0.27 *	0.82 ± 0.16
*Nppb/Ppia*	0.75 ± 0.12	1.14 ± 0.12 *	0. 82 ± 0.09	1.43 ± 0.12 *	0.73 ± 0.13 #
*Bax/Ppia*	0.89 ± 0.19	1.69 ± 0.34	0.97 ± 0.29	0.71 ± 0.08	0.68 ± 0.12
*Bcl2/Ppia*	0.91 ± 0.08	1.01 ± 0.14	0.83 ± 0.1	0.80 ± 0.05	0.80 ± 0.19
*Bax/Bcl2*	0.96 ± 0.18	1.63 ± 0.13 *	1.26 ± 0.26	0.87 ± 0.15 #	1.21 ± 0.34

Values are presented as mean ± S.E.M., * *p*  <  0.05 vs. control group, # *p*  <  0.05 vs. DOXO-only group (*n* = 5–9, one-way ANOVA, Holm-Sidak post hoc test). DOXO: doxorubicin, L: losartan, M: mirabegron, *Bax*: BCL2-associated X apoptosis regulator, *Bcl2*: B-Cell CLL/lymphoma 2 apoptosis regulator, *Mmp2*: matrix metalloprotease-2, *Mmp9*: matrix metalloprotease-9, *Nppa*: atrial natriuretic peptide A, *Nppb*: atrial natriuretic peptide B, *Ppia*: peptidylprolyl isomerase A (house-keeping gene for normalization).

**Table 6 ijms-23-02201-t006:** The effects of losartan, mirabegron, and their combination on left ventricular expression of redox enzymes at the mRNA level in DOXO-induced chronic cardiotoxicity at week 9.

Relative Gene Expression	Groups				
	Control	DOXO	DOXO + L	DOXO + M	DOXO + L + M
*Nos1/Ppia*	0.80 ± 0.17	0.68 ± 0.07	0.7 ± 0.16	0.64 ± 0.22	0.65 ± 0.05
*Nos2/Ppia*	0.48 ± 0.04	1.00 ± 0.17 *	0.75 ± 0.10 *	0.98 ± 0.2 *	0.64 ± 0.06 *
*Nox4/Ppia*	0.96 ± 0.15	1.24 ± 0.21	1.20 ± 0.39	0.24 ± 0.10 * ^#^	0.38 ± 0.11 ^#^
*Sod1/Ppia*	1.05 ± 0.05	0.97 ± 0.06	0.96 ± 0.13	0.78 ± 0.15	0.62 ± 0.09 *
*Sod2/Ppia*	0.94 ± 0.05	0.88 ± 0.06	0.80 ± 0.09	0.82 ± 0.09	0.51 ± 0.07 * ^#^
*Sod3/Ppia*	1.21 ± 0.15	1.09 ± 0.2	0.96 ± 0.21	0.54 ± 0.17 *	0.41 ± 0.09 * ^#^
*Cat/Ppia*	0.46 ± 0.04	0.43 ± 0.03	0.48 ± 0.08	0.37 ± 0.04	0.24 ± 0.04 * ^#^

Values are presented as mean ± S.E.M., * *p*  <  0.05 vs. control group, # *p*  <  0.05 vs. DOXO-only group (*n* = 5–9, ANOVA on ranks in the cases of *Nos1, Nos2, Nox4*, and *Sod2* expressions, and one-way ANOVA for the other parameters, Holm-Sidak post hoc test). DOXO: doxorubicin, L: losartan, M: mirabegron, *Cat*: catalase, *Nos1*: neuronal nitric oxide synthase, *Nos2*: inducible nitric oxide synthase, *Nox4*: NADPH-oxidase isoform 4, *Sod1*: superoxide dismutase 1 (cytoplasmic), *Sod2*: superoxide dismutase 2 (mitochondrial), *Sod3*: superoxide dismutase 3 (extracellular), *Ppia*: peptidylprolyl isomerase A was used as the house-keeping gene for normalization.

## Data Availability

The datasets used and/or analyzed during the current study are available from the corresponding authors on a reasonable request.

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
