# Peer review of "Investigation of the Antiremodeling Effects of Losartan, Mirabegron and Their Combination on the Development of Doxorubicin-Induced Chronic Cardiotoxicity in a Rat Model"

_ijms, 2022, doi:10.3390/ijms23042201_

Round 1

Reviewer 1 Report

This is a basic study, which aimed to investigate the potential anti-remodeling effects of the widely-used ARB losartan, theβ3AR agonist mirabegron, and their combination on the development of DOXO-induced chronic cardiotoxicity in a rat model. The authors concluded that the development of DOXO-induced systolic and diastolic dysfunction might be prevented or markedly slowed down by mirabegron and the combination of mirabegron plus losartan if their administration started in the early stages of DOXO-induced chronic cardiotoxicity without a severe decrease in LVEF. In this paper, only mirabegron prevented the development of DOXO-induced LV fibrosis in our model. The anti-remodeling effects of mirabegron seem to be independent of theβ3AR/eNOS mediated pathways in DOXO-induced chronic cardiotoxicity and probably associated with the repression on the SMAD2/3-mediated fibrotic pathway. Losartan failed to improve the severity of systolic dysfunction but had beneficial effects on diastolic dysfunction, SERCA2a level, and tissue inflammation in our DOXO-induced chronic cardiotoxicity model. This is an important issue, and this reviewer considers that the authors well performed the present study, reporting many results. This reviewer has some comments as described below.

Major comments:

  1. Page 4, lines 143-145. In this experiment, 1 in the DOXO only group, 2 in the losartan treated DOXO group, 4 in the mirabegron-treated DOXO group, and 2 in the losartan plus mirabegron-treated DOXO group died. How do the authors consider this issue? They should add the discussion.
  2. This paper has too many information. The authors should show the differential effects of losartan, mirabegron, and the combination as a summarized figure in the Discussion section.

Author Response

Reviewer #1

Comment 1.1.

This is a basic study, which aimed to investigate the potential anti-remodeling effects of the widely-used ARB losartan, the β3AR agonist mirabegron, and their combination on the development of DOXO-induced chronic cardiotoxicity in a rat model. The authors concluded that the development of DOXO-induced systolic and diastolic dysfunction might be prevented or markedly slowed down by mirabegron and the combination of mirabegron plus losartan if their administration started in the early stages of DOXO-induced chronic cardiotoxicity without a severe decrease in LVEF. In this paper, only mirabegron prevented the development of DOXO-induced LV fibrosis in their model. The anti-remodeling effects of mirabegron seem to be independent of theβ3AR/eNOS mediated pathways in DOXO-induced chronic cardiotoxicity and probably associated with the repression on the SMAD2/3-mediated fibrotic pathway. Losartan failed to improve the severity of systolic dysfunction but had beneficial effects on diastolic dysfunction, SERCA2a level, and tissue inflammation in this DOXO-induced chronic cardiotoxicity model. This is an important issue, and this reviewer considers that the authors well performed the present study, reporting many results. This reviewer has some comments as described below.

Answer 1.1.

We thank the Reviewer for the positive evaluation of our MS. We have addressed the valuable remarks of the Reviewer and hope that our answers and modifications indicated in red in the revised MS are satisfactory for the Reviewer now (see below).

Comment 1.2.

Page 4, lines 143-145. In this experiment, 1 in the DOXO only group, 2 in the losartan treated DOXO group, 4 in the mirabegron-treated DOXO group, and 2 in the losartan plus mirabegron-treated DOXO group died. How do the authors consider this issue? They should add the discussion.

Answer 1.2.

We thank the Reviewer for pointing out this critical issue. In the revised MS (page 5, lines 156-165), we added the time point of death of each animal in the DOXO groups in the Results section as follows:

“Altogether nine animals died in the DOXO groups (n=1 in the DOXO only group at week 4, n=2 in the losartan treated DOXO group [1 animal at week 4 before the start of losartan treatment and 1 animal 1 day before the termination at week 9], n=4 in the mirabegron-treated DOXO group [1 animal was excluded and terminated earlier due to its poor echocardiographic result and body weight at week 4, 1 animal before the start of mirabegron treatment due to abdominal ulceration at week 4 and 2 animals during the mirabegron treatment at weeks 7 and 8], and n=2 in the losartan plus mirabegron-treated DOXO group [1 animal was excluded and terminated earlier due to its poor echocardiographic results at week 4 and 1 animal during the losartan plus mirabegron treatment at week 6].”

The Discussion Section in the revised MS was also expanded according to the requests of Reviewer #1 and Reviewer #2 (pages 15-16, lines 561-574) as follows:

“In the present study, the mortality rate (Lódi et al. 2019), echocardiographic parameters, heart rate (Babaei et al. 2020; Baniahmad et al. 2020), blood pressure (Eisvand et al. 2021; Wu et al. 2018), serum lipid (Ikewuchi et al. 2021; Haybar et al. 2019), autopsy, and histologic findings in our DOXO-induced chronic cardiotoxicity model are consistent with the literature data on preclinical models (Podyacheva et al. 2021) and similar to those seen in tumor survivor patients treated with DOXO (McDonagh et al. 2021; Zamorano et al. 2016; Lenneman and Sawyer 2016). Among the nine animals that died in the DOXO groups, 5 rats died before the start of the treatments or should be excluded due to poor systolic function (LVEF<40%) assessed by echocardiography. During the treatments with losartan, mirabegron, and their combination, 4 animals died. These cases might be the consequences of DOXO treatment and not the side effects of the drugs administered in this study. To decide if the used drugs have severe side effects in DOXO-induced cardiotoxicity, more parameters, including arrhythmias, should be tested in more doses and follow-up time points with higher sample numbers in the groups”.

We hope that our modifications in the revised MS are satisfactory for the Reviewer.

Comment 1.3.

This paper has too much information. The authors should show the differential effects of losartan, mirabegron, and the combination as a summarized figure in the Discussion section.

Answer 1.3.

We thank the Reviewer for this valuable comment. According to the request of the Reviewer, a summary Figure is added to the Revised MS after the Conclusions Section (page 24).

Reviewer 2 Report

Doxorubicin treatment in cancer patients has been hampered due to a serious side effect on cardiac function. Therefore, tremendous efforts have been made to reduce this side effect using multiple drug treatments. However, it is an unsolved issue yet.

In this respect, Freiwan and colleagues have evaluated the therapeutic effects of ARB, Losartan, and/or b3AR agonist, Mirabegron in terms of systolic and diastolic cardiac dysfunction induced by DOXO treatment.

Overall, the concept and approaches are very straightforward, and all experiments seem well-executed, but some questions remain in this version of the manuscript.

Here are some comments.

Major comments

  1. The authors mentioned the number of animals that died during the entire procedure in the very first part of the result section. It seems that more numbers of rats died in the Losartan and Mirabegron-treated groups compared to DOXO only group. For example, in case of the Mirabegron-treated group, it showed about 40% mortality, because 4 out 10 rats died during the procedure. Although cardiac dysfunction is significantly reduced in surviving animals, what is the fundamental benefit of these treatments if the mortality is higher in treated groups?

  1. The authors have concluded that Losartan and the combinatory treatment ameliorated systolic dysfunction by showing the LVESD. However, IVCT is not significantly changed in combinatory treatment. In fact, it looks worse than individual treatment of Losartan or Mirabegron treatment based on Table 3.

How can you explain this discrepancy?

  1. In Table 4 and Fig. 4, heart weight is significantly reduced in DOXO-treated rats, but cardiomyocyte cross-sectional areas were not changed even though more fibrosis is observed. Did it happen because of apoptosis induced by DOXO in cardiomyocytes?

If this is the case, the authors need to show the data for apoptosis assay because heart weight and cardiomyocyte size are normally highly correlated, which is not observed in this model.  

  1. TGF-b-smad is a core signaling pathway in tissue fibrosis including the heart. Based on this, the authors have shown the enhanced expression levels of smad2 and 3in DOXO only treated group. Since only the Mirabegron-treated group downregulated smad 2 and 3 expression levels (Fig 5) as well as collagen deposition (Fig 4), the authors concluded that Mirabegron, but not Losartan, ameliorated LV fibrosis. However, phosphorylation of smad 2 and 3 is a major contributor activating the TGF-smad signaling pathway, thus phosphorylated forms of smad 2 and 3 levels should be measured. Also, it is recommended to show endogenous negative regulators of this pathway such as smad 6 or 7 to clarify the author’s conclusions although the authors stated that further mechanistic studies are needed in terms of the inhibition of the TGF-b/Smad pathway in the discussion section.

  1. In Fig 6, the western image is not clear, especially for SERCA2a. It seems too fuzzy to quantify the expression level. If it is doable, the reviewer highly recommends performing another western blotting for SERCA2a since it is such an important marker for cardiac dysfunction.

Minor comments

  1. The numbers in author affiliation are not matched. No authors are linked with No. 6 and 7.
